

# Missing sea-level rise in southeast Greenland during and since the Little Ice Age

Sarah A. Woodroffe[1], Leanne M. Wake[2], Kristian K. Kjeldsen[3], Natasha L.M. Barlow[4], Antony J.
Long[1], Kurt H. Kjær[5]
[1]Department of Geography, Durham University, Lower Mountjoy, South Road, Durham, DH1 3LE,
UK, s.a.woodroffe@durham.ac.uk
[2]Department of Geography and Environmental Sciences, Northumbria University, Ellison Place,
Newcastle upon Tyne, NE1 8ST, UK, leanne.wake@northumbria.ac.uk
[3]Geological Survey of Denmark and Greenland (GEUS), 1350 Copenhagen K, Denmark, kkk@geus.dk
[4]School of Earth and Environment, University of Leeds, LS2 9JT, UK, n.l.m.barlow@leeds.ac.uk
[5]GeoGenetics, Globe Institute, University of Copenhagen, 1350 Copenhagen K, Denmark,
kurtk@sund.ku.dk
*Correspondence to:* Sarah A. Woodroffe (s.a.woodroffe@durham.ac.uk)
**Abstract**
The Greenland Ice Sheet has been losing mass at an accelerating rate over the past two decades.
Understanding ice mass and glacier changes during the preceding several hundred years, prior to
geodetic measurements, is more difficult because evidence of past ice extent in many places was later
overridden. Saltmarshes provide the only continuous records of Relative Sea Level (RSL) from close
to the Greenland Ice Sheet that span the period of time during and since the Little Ice Age (LIA) and
can be used to reconstruct ice mass gain and loss over recent centuries. Saltmarsh sediments collected
at the mouth of Dronning Marie Dal, close to the Greenland Ice Sheet margin in southeast Greenland,
record RSL changes over the past c. 300 years through changing sediment and diatom stratigraphy.
These RSL changes record a combination of processes that are dominated by local/regional changes in





Greenland Ice Sheet mass balance during this critical period that spans the maximum of the LIA and
20th Century warming.  In the early part of the record (1725-1762 CE) the rate of RSL rise is higher
than reconstructed from the closest isolation basin at Timmiarmiut, but between 1762-1880 CE the RSL
rate is within the error range of rate of RSL change recorded in the isolation basin.  RSL begins to
slowly fall around 1880 CE and then accelerates since the 1990s, with a total amount of RSL fall of
$0.08 \pm 0.1$ m in the last 140 years.  Modelled RSL, which takes into account contributions from post-
LIA Greenland Ice Sheet Glacio-isostatic Adjustment (GIA), ongoing deglacial GIA, the global non-
ice sheet glacial melt fingerprint, contributions from thermosteric effects, the Antarctic mass loss sea-
level fingerprint and terrestrial water storage, over-predicts the amount of RSL fall since the end of the
LIA by at least 0.5 m.  The GIA signal caused by post-LIA Greenland Ice Sheet mass loss is by far the
largest contributor to this modelled RSL, and error in its calculation can have a large impact on RSL
predictions at Dronning Marie Dal.  We cannot reconcile the modelled RSL and the saltmarsh
observations, even when moving the termination of the LIA to 1800 CE and reducing the post-LIA
Greenland mass loss signal by 30 %, and a 'budget residual' of +~2.5 mm/yr since the end of the LIA
remains unexplained.

Keywords: Greenland, relative sea level, saltmarsh, glacio-isostatic adjustment, Little Ice Age, sea-level
budget

47        **1.      Introduction**

Studies using a range of different geodetic methods all agree that the Greenland ice sheet (GrIS) has
been losing mass at an accelerating rate over the past two decades (Bevis et al., 2019, 2012; Chen et al.,
2021; Khan et al., 2015; Moon et al., 2012; Pritchard et al., 2009; The IMBIE Team, 2020; van den
Broeke et al., 2009). There is however less known about when and at what rate ice mass loss occurred
in Greenland during the last millennium until the start of the satellite and GPS eras, during periods of
climate warming and cooling (Briner et al., 2020; Khan et al., 2020; Kjær et al., 2022).  Using Little Ice
Age (LIA) trimlines and stereo-photogrammetric imagery recorded between 1978-1987, Kjeldsen et al.
(2015) estimated an average Greenland-wide total ice mass loss of c. 75 Gt/yr during the 20th Century.
However, understanding how the rate of mass loss varied during the 20th Century is more complex



because it requires us to put a date on the end of the LIA, and to find a way of reconstructing mass loss
fluctuations without the help of continuous geodetic data. Understanding ice mass and glacier changes
during the preceding several hundred years is even more difficult because evidence of past ice sheet
extent in many places has been overridden by later advances (Briner et al., 2011; Kjær et al., 2022).

Salt marshes in nearfield settings record the timing and magnitude of fluctuations in ice mass during
the last few centuries through changes in relative sea-level (RSL) (e.g. Long et al., 2012). RSL reflects
the interplay of different cryosphere and oceanic processes but the dominant process close to an ice
sheet is the visco-elastic signature of local and regional mass changes through time (Farrell and Clark,
1976). Salt marshes form in the upper part of the intertidal zone and can continuously accumulate
organic sediment (Allen, 2000). Salt marshes in Greenland are generally small features with a very
short growing season, low sedimentation rates and may be affected by interactions with winter shore-
fast ice (Lepping and Daniëls, 2007). However, they can survive in these conditions and provide the
only continuous records of RSL from close to the GrIS that span the period during and since the LIA
and can be used to reconstruct ice mass gain and loss over recent centuries (Long et al., 2012, 2010;
Woodroffe and Long, 2009).

This study reports for the first time a continuous RSL record over the past ~300 years from a salt marsh
within 5 km of the ice sheet margin in southeast Greenland. The sediments and plant remains in the
marsh record RSL fluctuations over the last few hundred years and therefore provide a unique record
of changes in regional RSL during and since the LIA in Greenland. We predict local RSL changes by
creating a sea-level budget which includes predictions from a Glacio-Isostatic Adjustment (GIA) model
with c. 430 Gt ice mass loss in southeast Greenland between the end of the LIA and 2010 (as defined
by Kjeldsen et al., 2015), and estimates of other contributions since the end of the LIA including mass
loss from Greenland peripheral glaciers, non-Greenland ice, the thermosteric contribution and the effect
of terrestrial water storage in the 20[th] and 21[st] Centuries. Comparing the modelled sea-level budget and
the saltmarsh data provides an opportunity to consider potential errors in both methods and suggest how
we might bring model and data estimates closer together, as well as develop better understanding of the



nature of historical RSL in southeast Greenland and implications for coastline response to future,
enhanced GrIS and peripheral glacier melt.

**2. Study site and methods**
*2.1 Field site and glacial history of the region*
The saltmarsh record is from 63.470°N, -41.925°W at the head of Dronning Marie Dal in southeast
Greenland (Figure 1A,B, Fig 2). The saltmarsh is fed by freshwater and sediment from Dronning Marie
Dal, a formerly glaciated valley that drains part of the nearby Skinfaxe outlet glacier. Dronning Marie
Dal is at the head of the 50 km long marine fjord Søndre Skjoldungesund which together with Nørre
Skjoldungesund encompass the glaciated island of Skjoldungen (Figure 1C). The northern fjord has a
bedrock sill mid-fjord at c. 215 m below sea level, while the southern fjord has a narrow central section
with a sill located at 77 m below sea level (Kjeldsen et al., 2017). The narrow stretch connecting the
two fjords at their inland extent is generally shallow, sheltering the salt marsh at Dronning Marie Dal.
The region is dominated by long, steep-sided marine fjords with the GrIS ending at the coast in marine-
terminating outlet glaciers.

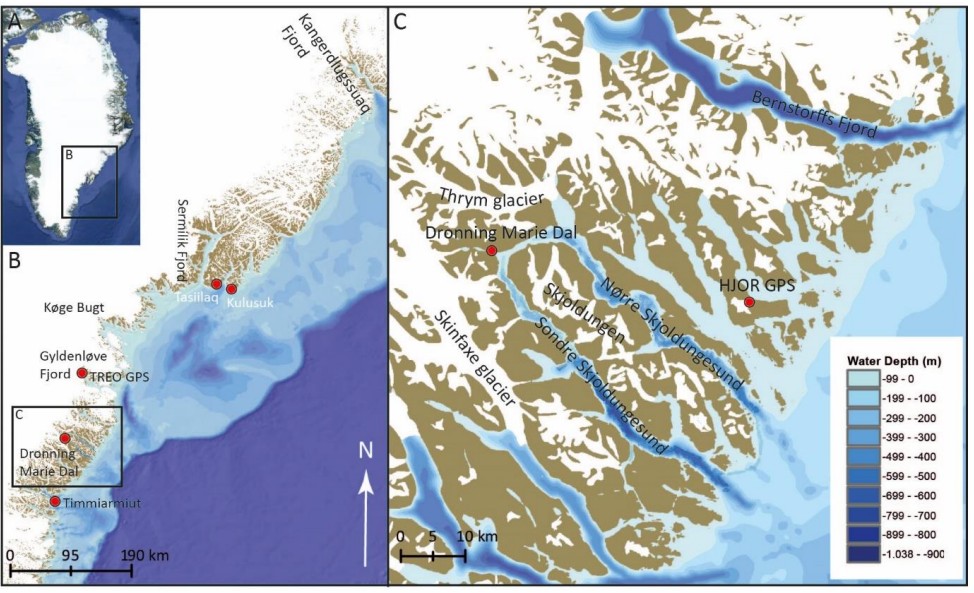



*Figure 1. A) Map of Greenland © Google Earth, B) Southeast Greenland region showing the location*
*of the field site (Dronning Marie Dal) alongside other studied fjords, C) Dronning Marie Dal saltmarsh*
*at the head of Sondre Skjoldungesund, between the Skinfaxe and Thrum glacier margins.*

Relatively little is known about the deglacial history of the southeast compared to the southwest of
Greenland. Most work has been undertaken in the large fjords (e.g. Kangerdlugssuaq, Sermilik, Køge
Bugt, Gyldenløve, Bernstorffs Fjord, Figure 1) to the north of the field area using [10]Be measurements
to reconstruct fjord deglaciation. During the LGM the ice sheet reached the shelf edge (50-80 km from
the outer coast) in this region and in the offshore Kangerdlugssuaq Trough to the north of the study area
the ice sheet started to retreat by c. 17 ka BP (Funder et al., 2011). Onshore deglaciation at the outer
coast occurred earlier to the north (Kangerdlugssuaq - 11.8 +/- 1ka BP) compared to the south
(Bernstorffs Fjord - 10.4 +/- 450 ka BP), driven by incursion of warm Atlantic water into the fjords
from the Irminger Current, moderated by local coastal bathymetry and atmospheric warming during the
early Holocene (Dyke et al., 2018, 2014; Hughes et al., 2012). [10]Be dates on boulders from outer and
inner Skjoldungesund suggest deglaciation here occurred in the early Holocene (inner fjord by 10.4
±0.4 ka BP) (Levy et al., 2020). Following retreat from the shelf edge, the deglaciation model HUY3
simulates retreat onshore by 10 ka BP, which largely agrees with the field evidence from
Skjoldungesund, with the ice sheet slightly inland of its LIA maximum position at 4 ka BP (Lecavalier
et al., 2014). The deglacial marine limit is low in this region (c. 20-40 m) suggesting less deglacial
mass loss compared to elsewhere in Greenland (Funder and Hansen, 1996). Observations of strandlines
up to 75 m above sea level in this region, reported by Vogt (1933) are cut into bedrock and are highly
unlikely to be of marine origin.

The HUY3 geophysical model predicts slight crustal subsidence at the coast today caused primarily by
a local late Holocene neoglacial readvance (resulting in RSL rise of 1-1.5 mm/yr over the last 1000
years) (Lecavalier et al., 2014). However, a recent GPS-derived GIA model (GNET-GIA) offers an
alternative solution with GIA uplift calculated at +2.8mm/yr and +3.1 mm/yr at nearby HJOR and
TREO GPS sites (Figure 1), which would result in pre-20[th] Century RSL fall at Dronning Marie Dal
(Khan et al., 2016). By comparing GPS data and absolute gravity observations over a 20-year period,



van Dam et al (2017) also suggest ongoing GIA uplift of +4.5 +/- 1.4 mm/yr at Kulusuk (300 km to the
north). These GIA estimates, based on modern observations, are corrected for elastic deformation in
response to modern mass balance changes to predict ongoing deglacial GIA. The most recent
examination of Greenland GIA model outputs and GPS data by Adhikari et al. (2021) suggests that
residual uplift caused by mass loss since the Medieval Warm Period, and in particular since the LIA,
accompanied by a reduced mantle viscosity on sub-centennial timescales, can explain the observed
discrepancy between uplift rates from HUY3 and elastic-corrected GPS uplift rates around Greenland.

LIA moraines are situated beyond the current frontal margins of the GrIS and local glaciers in this
region and demonstrate clearly that glacial retreat has occurred during the 20th Century (Bjork et al.,
2012). The instrumental temperature record from Tasiilaq indicates 2°C per decade of warming
between 1919 and 1932 CE (the early twentieth-century warming (ECW)), followed by cooling during
the 1950's to 1970's and steady temperature rise of 1.3°C per decade since 1993 (Bjork et al., 2012;
Chylek et al., 2006; Wood and Overland, 2010). Despite these decadal temperature fluctuations, and
the overall pattern of post-LIA retreat of southeast Greenland glaciers, the nearest glaciers to the field
site (Skinfaxe and Thrym, Figure 1C) have been relatively stable at their present positions since at least
the 1930s (Bjork et al., 2012). It is important to note however that Skinfaxe sits on a ledge in its fjord
system so would require significant thinning to dislodge it from its current position and Thrym Glacier
appears to be resting on shallow prograding bedrock (Bjork et al., 2012; Morlighem et al., 2017). The
total ice mass loss from the two drainage basins closest to the field site (Central East and South-East in
Kjeldsen et al., 2015) is 249 Gt between the end of the LIA and 1983, 134 Gt between 1983 and 2003
and 45 Gt between 2003 and 2010, based on the volume of loss from LIA trimlines and more recent air
photos. There is a significant increase (~70%) in the amount of regional mass loss during the post-1983
period compared to earlier in the 20[th] Century. We hypothesise that regional ice mass loss since the end
of the LIA should produce a visco-elastic GIA response recorded as variable 20th century RSL change
by local salt marsh sediments, such as those at Dronning Marie Dal (Figure 2).





*Figure 2. A) photograph looking East down the Dronning Marie Dal valley towards the head of Sondre*

*Skoldungesund and the salt marsh where the valley meets the fjord. B) photograph of the Dronning*

*Marie Dal salt marsh showing the low-angled relief of the marsh and zonation of salt marsh vegetation*

*(high marsh in the foreground).*

*2.2 Reconstructing RSL using saltmarsh sediments*

We collected salt marsh sediments by digging a small pit using a spade from the present-day high salt

marsh at the mouth of Dronning Marie Dal (Figure 1C, 2). The analysed sediment section is 13 cm

thick, with organic silt containing saltwater-tolerant diatoms situated over compacted sand-rich silt

where no diatoms are present (Figure 3). We sampled the fossil sediment section at 0.25 cm intervals

in the top 1 cm, and at 0.5 cm intervals further downcore to provide high-resolution RSL estimates,

bearing in mind the slow rate of sedimentation in most Greenlandic salt marshes (Long et al., 2012;

Woodroffe and Long, 2009). To reconstruct local RSL we investigated diatom assemblages across the





present-day salt marsh in the same location to understand changes in assemblages with elevation across
the upper part of the intertidal zone (Figure 3A). We then compared these assemblages to those found
through the sediment core using a visual assessment technique, that places weight on certain key taxa
that change abundance at clearly defined elevations (Long et al., 2012, 2010; Woodroffe and Long,
2009). The main species used to reconstruct RSL are the high marsh/freshwater species *Pinnularia*
*intermedia* and the high to low marsh species *Navicula cincta*. We prefer this method over a transfer
function approach (e.g. Barlow et al., 2013) because it relies on certain indicator species that occur at
narrowly defined levels, but also utilises other evidence such as vertical diatom succession and the
stratigraphy to interpret changes in RSL.

We initially calculated the elevations of modern and fossil saltmarsh samples to mean sea level (MSL)
using a high-precision dGPS. However due to technical issues with post-processing we instead rely on
tidal data from Timmiarmiut (100 km to the S) and tidal predictions from Tasiilaq (300 km to the NE)
collected during our fieldwork, along with knowledge about saltmarsh vegetation zonation in Greenland
and their general relationship to tidal levels, to relate fossil and modern saltmarsh elevations to mean
sea level (MSL). The tidal data from Timmiarmiut show that although the timing of daily tidal
fluctuations differs to predictions for Tasiilaq, the amplitude of tidal fluctuations is remarkably similar
(within 0.1 m). The tidal range (lowest to highest astronomical tide) at the outer coast is approximately
3.7 m. We have some confidence therefore that tidal predictions for Tasiilaq are applicable (with a time
correction) along the outer coast anywhere between Tasiilaq and Timmiarmiut, although the distances
involved are large. This leaves the issue of tidal range amplification or dampening in fjord-head settings
to consider, as the Dronning Marie Dal site is c. 50 km up-fjord from the open ocean (Figure 1C). This
is considered elsewhere in Greenland by Richter et al., (2011) who show that this effect is variable due
to fjord bathymetry and cross-section geometry, and ranges from -9 cm to +14 cm up fjord compared
to the fjord mouths on the west coast in fjords of similar length to Søndre Skoldungesund. Modern
saltmarsh vegetation at Dronning Marie Dal grows between 0.1 m above Highest Astronomical Tide
(HAT) and 0.08 m below Mean High Water of Spring Tide (MHWST) levels, which is very similar to
saltmarsh vegetation ranges we have observed elsewhere in southeast and southwest Greenland
(unpublished data and Woodroffe and Long, 2010, 2009). We are therefore confident that any effect of



the fjord-head setting on tidal range is small. We have not included an uncertainty estimate in our
overall RSL reconstruction to reflect this, because the uncertainty in the proxy elevations is already of
a similar magnitude (±0.10-0.15 m, see Table S2 in Supplementary Information).

*2.3 Chronology*
To provide a chronology to constrain the timing of reconstructed RSL changes we use a range of
complementary methods to maximise the precision of the resultant age-depth model. Very low
concentrations of $^{210}$Pb in the sediments required us to use other methods to provide recent
sedimentation rates. We investigated the presence of Total Mercury (Hg) (mg/kg, which includes both
mineral and atmospheric deposition) within the sediments using acid dissolution and quadrapole ICP-
MS as an indicator of anthropogenic emissions. Other studies in western and northern Greenland note
that between 1850-1900 CE there is more than a 2-fold increase in abundance of total Hg in lake
sediments compared to late Holocene levels (Bindler et al., 2001; Lindeberg et al., 2006; Shotyk et al.,
2003; Zheng, 2015), whereas Perez-Rodriguez et al. (2018) see a rapid increase in Hg abundance from
1880 onwards in southern Greenland. We therefore assume that the onset of detectable Hg above
background level in the Dronning Marie Dal saltmarsh sediments at 4-4.5 cm indicates an age of 1850-
1900 CE and use 1875 ± 25 CE in the age-depth modelling described below. For the earlier part of the
sediment record we submitted seeds and leaves from saltmarsh and nearby freshwater plants picked
from multiple horizons within the sediment for AMS $^{14}$C dating at the $^{14}$Chrono centre at Queen's
University, Belfast (Table 1). We generated an age-depth model for the whole sequence using the
*P_Sequence* approach with *variable k* in Oxcal v. 4.3 using the IntCal20 calibration curve (Bronk
Ramsey, 2009; Ramsey and Lee, 2013; Reimer et al., 2020). The resultant age-depth model uses the
Hg chronohorizon (1850-1900 CE) and three $^{14}$C dates from lower in the sequence to estimate the age
of every 0.25 cm of sediment in the sediment section with associated uncertainty (Table 1 and Table S2
in supplementary information). The chronological uncertainty reported throughout this study is the
95% probability distribution (Bronk Ramsey, 2009).

We exclude the $^{14}$C ages at 6-6.5 cm (UBA28477) and 9-9.5 cm (UBA28478) from the age-depth model
because they were on extremely small samples (<0.8 mg carbon) and are from samples that mix seeds



and leaves from high salt marsh with freshwater plants that would not have been growing close together
at the time (based on the palaeoenvironment recorded by the fossil diatom assemblage, and the
distribution of diatoms and vegetation types on the present-day saltmarsh) (Table 1).  The dated
macrofossils from lower in the sequence are more likely to be autochthonous as the diatoms record a
high marsh to freshwater environment, close to HAT, at the time of deposition.

| Core depth (cm) | Lab number | $^{14}$C age (yr CE/BP) | $^{14}$C age error (yr/1 sigma) | F$^{14}$C | F$^{14}$C error | Cal curve | Dated material | Used in age model? |
|---|---|---|---|---|---|---|---|---|
| 6-6.5 | UBA28477 | 496 CE | 508 | 1.0265 | 0.1421 | | Carex subspathacea seeds/Empetrum nigrum leaves | N |
| 9-9.5 | UBA28478 | 1955 CE | 1 | 1.0107 | 0.0052 | | Carex subspathacea seeds/Empetrum nigrum leaves | N |
| 10-10.5 | UBA28481 | 208 BP | 67 | n/a | n/a | INTCAL20 | Carex subspathacea seeds | Y |
| 11.5-12 | UBA28476 | 134 BP | 93 | n/a | n/a | INTCAL20 | Carex subspathacea seeds | Y |
| 12-13 | UBA28479 | 44 BP | 45 | 0.99453 | 0.00555 | INTCAL20 + NHZ1 | Carex subspathacea seeds | Y |

*Table 1. Radiocarbon dated samples from the Dronning Marie Dal saltmarsh core.*

*2.4  Modelling RSL*
2.4.1 Deglacial RSL change
There is a high degree of uncertainty on the rate of GIA in south-east Greenland, owing largely to the
lack of Holocene RSL data points to constrain deglacial history. Marine ingression into an isolation
basin at Timmiarmiut (100 km SW of Dronning Marie Dal) at c. 1140 CE (Table 2, also see Figures
S1, S2 and Table S1 in the supplementary information) gives an empirical estimate of regional GIA and
suggests that the linear rate of background RSL change over the past millennium is in the range of +0.2
to +0.8mm/yr (Table 2).  We therefore use a mid-point value of +0.5 mm/yr as the rate of RSL change
due to ongoing deglacial GIA in this study, rather than model predictions outlined in Section 2.1 which
are not validated using RSL data from this region.



*Table 2. Isolation basin sea-level index point from Timmiarmiut used to calculate the rate of RSL due*
*to ongoing GIA in this study.*

| Location (lat,lon) | Sill height (m MTL) | Reference Water Level | RSL (m) | Max cal age CE | Min cal age CE | Cal age error +/- | $^{14}$C age | Lab code |
|---|---|---|---|---|---|---|---|---|
| Timmiarmiut XC1403A (62.4987, -42.2577) | 1.33 +/- 0.5 | Ingression (MHWST to HAT) | -0.24 +/- 0.5 | 1044 | 1243 | 99.5 | 873 +/- 30 | AAR 25631 |


2.4.2 Post Little Ice Age Greenland contribution
The post-LIA contribution to RSL at Dronning Marie Dal is computed using the sea level algorithm of
Kendall et al. (2005) computerised by Milne and Mitrovica (2003). This code computes the geoidal and
crustal response to ice and ocean loads on a spherically-symmetric Earth discretized into 25 km-thick
elastic layers as defined by Dziewonski and Anderson (1981), and three viscous layers comprising a
lithosphere, upper and lower mantle. Lithospheric thicknesses (L) in the range 71-120 km are
considered, with upper mantle ($v_{UM}$) and lower mantle ($v_{LM}$) viscosities of 0.1-1 x $10^{21}$ and 1-50 x $10^{21}$
Pa s explored to quantify the effect on predicted RSL change of different assumptions about Earth
viscosity structure.  The post-LIA ice history for the GrIS is derived from Kjeldsen et al. (2015) who
used a collection of aerial imagery from 1978-1987 CE to compare to historical trimlines assumed to
be indicative of a maximum LIA position of the ice sheet and use 1900 CE as a Greenland-wide year
of retreat from the maximum position, while acknowledging considerable local and regional
differences. The extrapolation method of point-scale changes in ice thickness over this time period to
the rest of the Greenland Ice Sheet is detailed in the methods section of Kjeldsen et al. (2015).

2.4.3 Contribution from Greenland glaciers
Changes in ice thickness in peripheral Greenland glaciers is determined in exactly the same way as the
post-LIA Greenland contribution. The peripheral Greenland glacier mass balance history is extracted
from Marzeion et al. (2015) and considered separately from the global glacier dataset (Section 2.4.4)
due to their proximity to the field site; the RSL response is computed as described in Section 2.4.2.

2.4.4 Contribution from global glaciers



We calculate the sea level contribution from global glaciers by first computing the global fingerprint
for a +1mm/yr barystatic contribution from glacier complexes defined in Marzeion et al. (2015, 2012)
since 1902. For the purposes of this calculation, we distribute the mass change across the glacierised
regions equally since the use of a 512 harmonic truncation masks sub 100 km-scale variability in ice
thickness change across regions outside of Greenland.  Ice thickness change will vary internally to each
glacierised area, but the great distance between southeast Greenland and many of the sources of melt
means that the solution is insensitive to spatially inhomogeneous changes in ice thickness within the
source regions.  Ice thickness changes for each of the global glacier complexes are discretized into
decadal loading intervals and the global sea level response is computed using the density configuration
in the Preliminary Reference Earth Model (PREM) (Dziewonski and Anderson, 1981). We use a
lithospheric thickness of 96 km to represent a global average applied to all glacial sites and omit the
viscous component from this calculation.  Dronning Marie Dal is proximal to glacier sources in Iceland
and Baffin Bay so should display some level of sensitivity to ice loss distribution over these glacierised
areas. However, it is in the 'near field' with respect to both of these sites, and therefore the use of a
more realistic ice loss distribution in these areas (e.g. peripheral thinning) will reduce the relative sea-
level rise recorded in southeast Greenland. The influence of low-latitude glaciers is excluded from the
sea level fingerprint calculations, as the areas of mass loss are below the spatial resolution of the
fingerprinting code. This simplified method produces similar results that of Frederikse et al. (2020).

2.4.5 Contribution from the Antarctic Ice Sheet
Loss of ice mass from either East or West Antarctic Ice Sheets will produce a relatively uniform sea-
level change fingerprint over the northern Hemisphere (Bamber and Riva, 2010; Mitrovica et al., 2001).
Recent Antarctic Ice Sheet change (1992-present) is relatively well-documented and quantified
(Meredith et al., 2019) compared to the period represented by the RSL data in this study.  However, a
recent study by Frederikse et al. (2020) that applied a Monte Carlo approach to balance the budget of
global sea-level rise since 1900 used estimates of 20[th] century Antarctic Ice Sheet mass balance obtained
from Adhikari et al. (2018) where the focus of mass loss throughout the 20[th] century is thought to be in
the West Antarctic Ice Sheet, amounting to a global sea-level change of 0.05 ±0.04 mm/yr. We use the





resulting ensemble from Frederikse et al's (2020) analysis to compute Antarctic Ice Sheet contribution
at Dronning Marie Dal.

2.4.6 Contribution from steric changes
To compute the contribution from salinity and temperature changes in the nearby ocean, the
Thermodynamic Equation of Sea Water (McDougall and Barker, 2011) (algorithm available here:
https://www.teos-10.org/) was applied to compute the steric height of the ocean. This uses a suite of
proximal monthly temperature-depth and salinity-depth profiles extracted from the CMIP6 database for
the 'historical' experiments covering the period 1850-2014. The 'historical' experiment was chosen to
produce timeseries of depth-dependant potential temperature and salinity because the experiment forms
part of the principal set of CMIP6 simulations, and the forcing datasets provided to the AOGCMs are
consistent with a set of atmospheric and ocean observations (Eyring et al., 2016). We use only one
configuration of the variant ID, which relates to initialisation time and procedure, specific model
physics and forcing (r1i1p1f1) across all AOGCMs considered (NASA-GISS-E2, CESM2, AWI,
CanESM5 and FGOALS). The model output from the CMIP6 database has a spatial resolution in the
range of 50-200 km, so we use profiles located within 300 km of Dronning Marie Dal to calculate an
average trend in steric height for the nearby ocean. The steric heights are computed to reference depth
levels of 500 m, 1000 m, 2000 m and 3000 m. Computing steric heights to different reference levels
allow us to determine which depth(s) in the ocean are contributing to steric height variability. Ivchenko
et al. (2008) determined that for the North Atlantic for the period 1996-2006, applying a reference level
of 1000-1500 m was sufficient to capture steric height variability, although this study provides trends
in steric height across the maximum depth level available by each model in the region proximal to
Dronning Marie Dal.

2.4.7 Terrestrial Water Storage
To estimate the contribution of changes in terrestrial water storage we utilise the ensemble of timeseries
of Frederikse et al. (2020) covering the time-period 1900-2018 CE. This dataset was compiled by
including the effects of natural variability in water reservoirs attributed to hemispheric-scale
atmospheric and ocean circulation changes (Humphrey and Gudmundsson, 2019), changes in storage



from dam building (Chao et al., 2008) and groundwater depletion activities (Döll et al., 2014; Wada et
al., 2016).

In the next section the results from the field work, RSL reconstruction and sea-level modelling are then
compared to better understand changes in mass balance and RSL over recent centuries in southeast
Greenland.

**3. Results**
*3.1 Modern diatom assemblages*
Diatoms are zoned by elevation across the upper part of the intertidal zone at Dronning Marie Dal, with
individual species providing useful information for reconstructing RSL. Above 2.2 m MTL (>0.34 m
above HAT) no diatoms were found in surface sediments, probably because the environment is too arid.
There is a distinctive assemblage containing *Pinnularia intermedia* (>10 % at HAT, increasing to ~55
% in the highest samples) which ends at 2.2 m MTL. We use this as a proxy sea level indicator and to
reconstruct palaeo-marsh surface elevation changes when we find >10 % of *Pinnularia intermedia* in
fossil counts (Figures 3A and B and Table S2 in supplementary information). We find this upper
intertidal/supratidal *Pinnularia intermedia* assemblage at every marsh we have studied in southeast and
southwest Greenland and use it to reconstruct RSL rather than using a transfer function approach as its
precision is as good as or better (Pinnularia intermedia is present in >15 marshes between 59° and 69°
N in southwest and southeast Greenland with a vertical range of 0.2-0.4 m; unpublished data and Long
et al., 2012, 2010; Woodroffe and Long, 2010, 2009). Where *Pinnularia intermedia* is <10 % in fossil
sample assemblages we consider other species, particularly *Navicula cincta* which is present in greatest
abundance in lower parts of the marsh, to reconstruct RSL (Table S2 in supplementary information).

*3.2 Core stratigraphy and biostratigraphy*
The core stratigraphy consists of a compacted basal freshwater organic silt-clay, grading upwards into
organic high saltmarsh sediments, and then into a slightly silt-rich organic low salt marsh towards the
surface, with an increase in LOI values in the top 2 cm (Figure 3B). Diatoms are well preserved in the
core and show a trend of falling palaeo-marsh surface elevation upwards from the base of the sequence



as *Pinnularia intermedia* declines and *Navicula cincta* increases in abundance.  In the top 3 cm
*Pinnularia intermedia* increases in abundance recording RSL beginning to fall and palaeo-marsh
surface elevation increasing (Figure 3B).

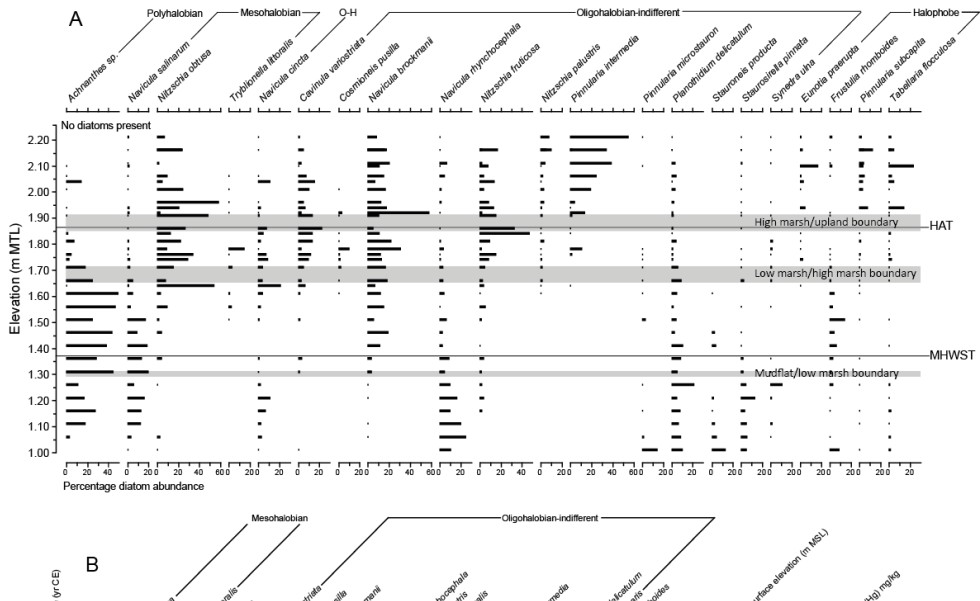

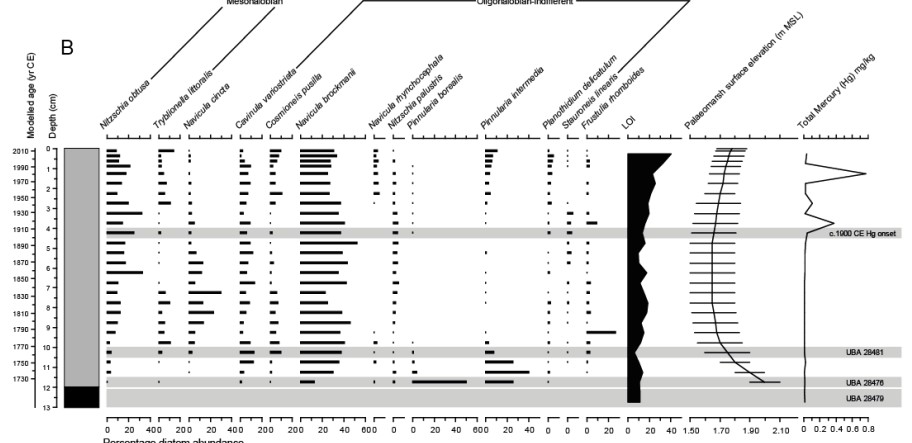


*Figure 3. A) Modern diatom data from the marsh at Dronning Marie Dal.  Data are expressed as %*
*total diatom valves (%TDV).  Only data >10% TDV are shown.  B) Fossil diatom counts, palaeo-marsh*
*surface elevation reconstruction and total Mercury measurements from the Dronning Marie Dal*
*saltmarsh core.  Diatoms are expressed as a %TDV and only taxa with >10% TDV are shown.*
*Stratigraphy is show in lefthand box where grey = saltmarsh sediment, black = freshwater peat.  Total*
*Mercury (mg/kg) was measured on salt marsh sediment using quadrapole ICP-MS.*





*3.3 RSL reconstructions*
The saltmarsh sediments and diatoms indicate long term RSL rise.  The rate of RSL rise at the start of
the record (+8-13 mm/yr between 1725-1762 CE; Figures 4B and C) is significantly higher than the
rate reconstructed from the closest isolation basin at Timmiarmiut (+0.2-0.8 mm/yr; Table 2).  This may
be due to LIA ice growth, including the glacier arm delivering sediment-laden meltwater to Dronning
Marie Dal, causing local ice loading and rapid infilling of accommodation space and salt marsh
development.  The rate of RSL rise declines rapidly over the period 1762-1880 CE and is within the
error range of the isolation basin rate during most of this period (+0.3-3 mm/yr).  This trend of rapid
and then slowly rising RSL between 1725-1880 CE is likely due to changes in the local LIA ice load
over this time period combined with ongoing millennial-scale GIA.  The HUY3 model predicts +1.44
mm/yr of RSL rise over the past 1000 years in this region (Lecavalier *et al.* 2014) which is the same
sign as the salt marsh and isolation basin RSL data during this period.  Other recent estimates of
centennial-scale GIA (Khan et al., 2016; van Dam et al., 2017) suggest that RSL should have been
falling over the past few hundred years at Dronning Marie Dal.  The isolation basin and salt marsh data
instead suggest that RSL was rising or close to stable from c. 1100 CE until c. 1880 CE.

Since 1880 CE RSL began to fall, which is indicated clearly in the diatom record by the reintroduction
and increasing abundance up core of *Pinnularia intermedia*, a high marsh diatom species (Figure 3B).
This high marsh environment gives the most precise RSL reconstructions and therefore we have the
most confidence in the pattern of RSL during this recent part of the record.  RSL began to fall around
1880 CE at a relatively constant rate (< -1mm/yr) until the 1990s, when the rate of fall accelerates to
between -1 and -3 mm/yr to present (2014 CE), resulting in a total amount of RSL fall of ~0.08 ±0.1 m
since 1880 CE (Figures 4B and C, Table 3, Table S2 in supplementary information).





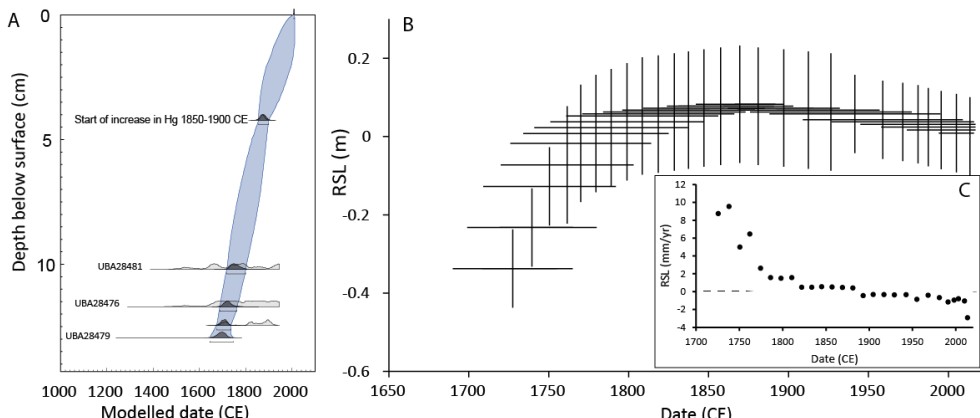


*Figure 4A) age-depth model using three $^{14}C$ ages and the Hg chrono-horizon, B) Dronning Marie Dal*

*RSL curve, C) rates of RSL change through time inferred from the RSL and age data.*


*3.4  Modelled RSL changes*
Published calculations of post LIA Greenland mass loss and other RSL contributors start at 1900 CE
(e.g. Kjeldsen et al., 2015; Marzeion et al., 2015), so we focus on this part of the salt marsh RSL record
to compare the reconstructed RSL with a modelled sea-level budget.  The different contributions to the
sea-level budget are summarised in Table 3 and Figure 5.  For an average Earth model configuration of
$L = 96km$, $\nu_{UM} = 0.5 \times 10^{21}$ Pa s and $\nu_{LM} = 10 \times 10^{21}$ Pa s, post-LIA ice mass loss (from the GrIS only)
resulted in sea level change of –5.9 mm/yr at Dronning Marie Dal between 1900-2010.  Between 1983-
2010 the modelled RSL rate was –10.1 mm/yr. Any chosen Earth configuration within the parameter
range explored does not significantly affect the predicted sea-level change; for 1900-2010, the range of
RSL fall was between –6.7 to –5.8mm/yr and 1983-2010 between -11.7 to -9.9 mm/yr.  The contribution
of peripheral Greenland glaciers to RSL was on average -1.7 ± 0.2 mm/yr between 1903 and present
day; with decadal-scale contributions of -3 to -5 mm/yr between 1923 and 1943.  Global glacier mass
loss contributes +0.24 ± 0.06 mm/yr RSL rise between 1903-2009.  Antarctica has contributed more
significantly to sea-level change in recent years; for the period 1992 to 2016, the Antarctic Peninsula
and the West Antarctic Ice Sheet are thought to have resulted in +0.06 ± 0.73 mm/yr of barystatic sea-
level change (Meredith et al., 2019). However, for the period 1850-2014 Frederikse et al. (2020)
compute +0.08 ± 0.08 mm/yr, rising to +0.2mm/yr between 1970-2018.





The range of values for the modelled steric contribution are in Table 4. They represent an upper estimate
of the magnitude and range of the steric component as only profiles showing significant RSL trends are
used when calculating the mean. From 1850-2014, trends in steric height are in the range –0.23 to
+0.18 mm/yr for a reference depth level of 1000 m and -0.36 to +0.28 mm/yr over a depth range of
2000 m. An observation-based analysis of trends in steric height by Frederikse et al. (2020) shows the
steric contribution from the upper 2000m of the ocean close to Dronning Marie Dal between 1957-2018
is +0.13 mm/yr (we include steric trends derived for the period 1950-2014 in Table 4 for comparison).
All models considered in Table 4 have larger values than Frederikse et al. (2020)'s estimates. Finally,
the impact of terrestrial water storage amounts to a sea level fall of –0.13 ± 0.06 mm/yr at Dronning
Marie Dal over the 20th century.

The different contributions to RSL are summed and plotted alongside the saltmarsh RSL data in Figure
5. The sum of components predicts RSL fall of between 0.58-0.93 m since 1900 CE. This prediction
is dominated by the contribution of GIA caused by post LIA Greenland and peripheral glacier mass
loss, which is only counteracted a little by the other components which mostly predict small amounts
of RSL rise. The saltmarsh data only reconstruct ~0.08 ±0.1 m of RSL fall since 1880 CE producing a
large mismatch between the sea-level budget and the saltmarsh RSL data. However the RSL data does
show an acceleration in the rate of RSL fall since the 1990s which agrees with accelerated GrIS mass
loss in the 1983-2003 and 2003-2010 time periods in Kjeldsen et al. (2015).

*Table 3. Calculated amounts and rates of RSL change from the various contributors to the RSL budget*
*at Dronning Marie Dal. Rates of RSL change are supplied with ± 2-sigma uncertainty unless specified:*
*\* uncertainty reflects assumed ± 10% error on rates which is larger than ± 2-sigma. \*\* steric sea level*
*contribution calculated from the average of significant trends for the 0-2000m depth interval from three*
*models in Table 4. \*\*\* GIA from nearby isolation basin ingression with uncertainty calculated from*
*upper and lower elevation reconstruction uncertainties.*

| Contribution to sea-level budget | Local or global | Time period | Contribution to RSL change (mm), upper and lower estimates | Rate of RSL change (mm/yr) assumed for common period of 1900-2012 |
|---|---|---|---|---|
|  |  |  |  |  |





| | | | calculated for common period of 1900-2012 | |
|---|---|---|---|---|
| GIA caused by post LIA ice mass loss* | Local | 1900-2010 | -724, -593 | -5.9 ± 0.6 |
| GIA caused by Greenland peripheral glacier mass change* | Local | 1903-2012 | -202, -166 | -1.7 ± 0.2 |
| Millennial-scale deglacial GIA*** | Local | 1900-2018 | 33, 88 | +0.5 ± 0.3 |
| **Local total** | | | | **-7.1 ± 0.6 mm/yr** |
| Global glaciers | Global | 1903-2012 | 20, 33 | +0.24 ± 0.06 |
| Antarctica | Global | 1900-2018 | 0, 18 | +0.08 ± 0.08 |
| Steric** | Global | 1850-2014 | -39, 39 | +0.00 ± 0.35 |
| Terrestrial water storage | Global | 1900-2018 | -21, -8 | -0.13 ± 0.06 |
| **Global total** | | | | **+0.19 ± 0.35 mm/yr** |
| Total modelled RSL change at Dronning Marie Dal 1900-2012 (see Figure 5) | | | -933, -589 | -6.9 ± 1.5 mm/yr |
| Rate of RSL change from saltmarsh data (1880-2014) | | | | -0.67 ± 1.7 mm/yr |

*Table 4: Mean trends in steric height anomalies for three reference levels (500, 1000 and 2000m) calculated from profiles within 300km of Dronning Marie Dal using five models participating in the CMIP6 analysis. In all cases, experiment variant ID was r1i1p1f1. Numbers in brackets denote number of profiles displaying significant trends in steric height from which the mean and 2-sigma trends were calculated. The AWI model produced no significant trends for either time-period whilst GISS-E2 did not produce significant trends for 1850-2014.*

| Model ID | Resolution (space) | Resolution (time) | 0-500m | 0-1000m | 0-2000m |
|---|---|---|---|---|---|
| **1850-2014** | | | | | |
| GISS-E2 | 200km | Monthly | - | - | - |
| CESM2 | 100km | Monthly | 0.08 ± 0.01 (34) | 0.17 ± 0.01 (26) | 0.09 ± 0.01 (7) |
| FGOALS | 100km | Monthly | 0.14 ± 0.12 (7) | 0.18 ± 0.16 (11) | 0.28 ± 0.04 (6) |
| AWI | 25km | Decadal | - | - | - |
| CanESM5 | 100km | Monthly | -0.12 ± 0.1 (13) | -0.23 ± 0.08 (13) | -0.36 ± 0.06 (13) |
| **1950-2014** | | | | | |
| GISS_E2 | 200km | Monthly | 0.17 ± 0.02 (10) | 0.36 ± 0.03 (6) | 0.75 ± 0.05 (3) |
| CESM2 | 100km | Monthly | 0.63 ± 0.15 (37) | 1.3 ± 0.11 (26) | 1.24 ± 0.07 (7) |
| FGOALS | 100km | Monthly | 0.43 ± 0.17 (11) | 0.57 ± 0.18 (12) | 0.81 ± 0.15 (6) |
| AWI | 25km | Decadal | - | - | - |
| CanESM5 | 100km | Monthly | 0.97 ± 0.34 (15) | 1.1 ± 0.32 (13) | 0.96 ± 0.2 (8) |



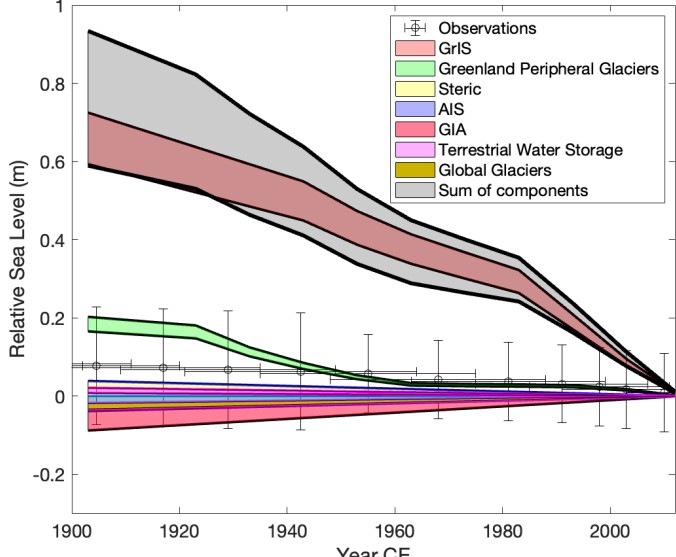

469

*Figure 5: Observed and modelled relative sea level change from 1900-2010 CE as a function of recent*

*and late Holocene Greenland ice thickness changes (GIA caused by the GrIS, Greenland peripheral*

*glaciers and millennial-scale GIA; the 'local' signal) and from sources outside of Greenland (steric*

*signal, AIS, terrestrial water storage and global glaciers). The sum of the modelled components is*

*shown as the grey shaded area and the GrIS and peripheral glacier contributions are shown with an*

*estimated ±10% uncertainty. The black crosses are the salt marsh-based RSL reconstruction.*

476

**4. Discussion**

The dominant contributors to post-LIA RSL change at Dronning Marie Dal are the adjustment of the

solid Earth and changes in geoid height in response to both post-LIA and millennial-scale Greenland

ice sheet changes. These contributors (ongoing GIA from the last deglaciation, post LIA Greenland

mass balance and mass loss from peripheral Greenland glaciers) amount to a modelled sea-level fall of

-7.1 mm/yr between 1900-2010. By contrast, the RSL contributors unrelated to cryospheric change in

Greenland only amount to modelled sea-level rise of +0.19 mm/yr, giving a total RSL fall of -6.9 mm/yr

between the end of the LIA and present (Table 3). This clearly does not fit with the observations from

the salt marsh data (Figures 4B, 5), which suggests that the rate of RSL fall between 1900-2013 is –

0.67 ± 1.7 mm/yr.






*4.1 Timing of the end of the LIA and Greenland ice sheet and peripheral glacier contribution*

To try to bring the post-LIA sea-level budget closer to the salt marsh observations, we explore two
possible sources of uncertainty in the dominant post LIA Greenland signal: 1) timing of the start of
post-LIA mass loss in Greenland and 2) greater uncertainty in modelled sea level associated with post-
LIA GrIS and peripheral glacier mass loss.

To explore the possibility that the total post-LIA Greenland mass loss occurred over a longer time period
we create four scenarios where the LIA maximum ice termination in Greenland is adjusted to begin at
1750, 1800, 1850 and 1900 CE, and the rate of mass loss is scaled accordingly with the end point
remaining at 2010 (as in Kjeldsen et al., 2015). We know that the LIA ice sheet response was different
around Greenland with multiple advance phases forced by different driving mechanisms, and it is
simplistic to suggest that the whole of the ice sheet began to lose mass simultaneously at 1900 CE (Kjær
et al., 2022), albeit it may serve as a Greenland-wide year. By adjusting the LIA termination date (and
therefore the start of Greenland and peripheral glacier mass loss) we can investigate the impact of earlier
ice retreat on RSL at Dronning Marie Dal. In this sensitivity analysis we recognise that moving the
LIA termination date in our modelling means that we are assuming the LIA ended simultaneously earlier
around the whole of Greenland, which is no more nuanced than assuming LIA termination at 1900 CE.
We also note that the glaciers closest to Dronning Marie Dal appear to have been at their LIA maximum
position in the early 20[th] Century, which does not agree with an earlier LIA end in this location (Bjork
et al., 2012), and a recent alkenone-based sea-surface temperature reconstruction from Nørre
Skjoldungesund suggests considerable warming here occurred late, between c. 1915-1945 CE
(Wangner et al., 2020). The analysis does however allow a first-order investigation into the sensitivity
of modelled post-LIA sea level to the length of time over which the post-LIA mass loss occurred.

The second parameter that we vary as part of this sensitivity study is the total amount of post-LIA mass
loss from the GrIS and peripheral glaciers, by assuming an error of up to -30% on these calculations.
Kjeldsen et al. 2015 report uncertainties in their mass loss estimates for the Southeast sector of the ice



sheet between 7-15 %, and so this sensitivity analysis allows us to test the effect on RSL at Dronning
Marie Dal of a smaller amount of mass loss since the end of the LIA in this region.

Varying both LIA termination date and total post LIA mass loss from the GrIS and peripheral glaciers
has a large effect on how much sea-level change from other components is required to close the post-
LIA budget (Figure 6). The 'budget residual' in Figure 6 refers to the misfit in mm/yr between the RSL
change reconstructed by the saltmarsh data and RSL change predicted by the sea-level budget
calculations. In essence this is the amount of sea-level change that we still need to 'find' to close the
budget even after we modify the timing and total amount of mass loss from the dominant contributors
to RSL change of GrIS and peripheral glacier retreat since the end of the LIA.

The time-period over which post LIA mass loss occurs is important for understanding the degree of
volume mismatch between the RSL observations and modelled contributions from the maximum extent
to present. Figure 6a indicates that moving the LIA termination date from 1900 CE to 1750 CE reduces
the 'budget residual' required to fit the RSL data from ~+5.5 to ~+3.5 mm/yr. This residual is reduced
further (to ~+2.5 mm/yr) when considered alongside a 30% reduction in the amount of mass loss from
the GrIS and peripheral glaciers compared to volumes reconstructed by Kjeldsen et al. 2015 (GrIS) and
Marzeion et al. 2015 (peripheral glaciers) (Figure 6b). Figures 6c and d further illustrate these results.
Figure 6c, where there is no reduction in the amount of post-LIA mass loss shows a good fit to the RSL
data when the LIA termination is moved to 1800 CE, but there remains a +4 mm/yr 'budget residual'
which must be accounted for from other parts of the sea-level budget. In Figure 6d, a similar good fit
to the RSL data is possible with a 30 % reduction in Greenland and peripheral glacier mass-loss and
LIA termination at 1800 CE. The remaining 'budget residual' is +2.5 mm/yr which again must be
accounted for from other parts of the sea-level budget.

The smallest calculated 'budget residual' (~+2.5 mm/yr) has to be found from processes causing sea-
level rise in southeast Greenland, such as millennial-scale Greenland GIA, Antarctic Ice Sheet melt, the
steric effect, and global glacier melt. The modelled sea-level budget suggests that these processes are
only small contributors to total sea-level change, with the sum of sources from outside Greenland only




+0.19 mm/yr since 1900 CE. The steric effect has the largest uncertainty, which we consider in Section
4.8 alongside other potential sources of error in our calculations. It is difficult however to see how the
contributors to RSL rise in southeast Greenland could be significantly larger before 1900 CE given the
cooler regional temperatures of the LIA.

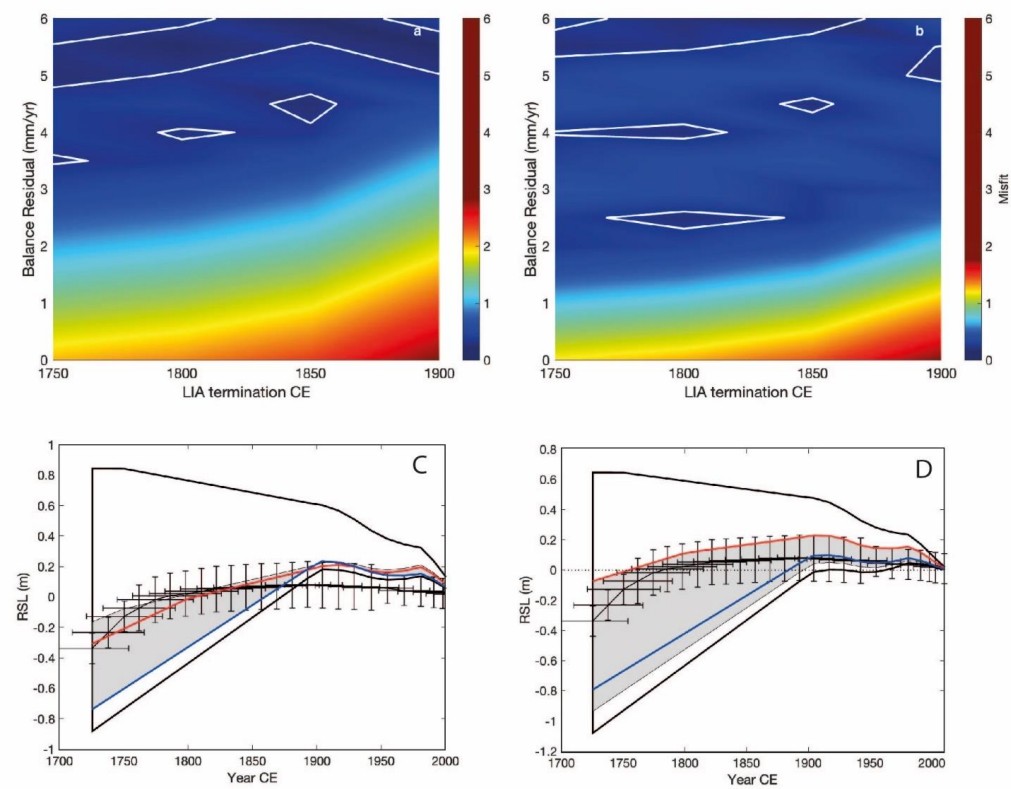


*Figure 6. a, b) Misfit plots showing model data-fit where combinations of 'budget residuals' and LIA*
*termination dates are considered with (a) no assumed error in the RSL contribution from the GrIS and*
*(b) a 30 % reduction in magnitude of sea-level change associated with local changes in the GrIS. Areas*
*within the white lines have a statistically equivalent fit to the RSL data, c) Modelled RSL from all*
*combinations of LIA termination date and budget residual, assuming no error in the RSL contribution*
*from the GrIS. Area within the black line denotes all possible combinations of RSL trends from LIA*
*terminations from 1750-1900 CE and budget residual rates between 0-6mm/yr. Grey shaded area*
*corresponds to RSL trends from within white lines on Figure 6a, demonstrating a statistically equivalent*



*fit to the data. For illustrative purposes, the red line denotes a modelled RSL scenario with a budget*
*residual rate of +4mm/yr and LIA termination date of 1800 CE; the blue line a modelled RSL scenario*
*with a budget residual rate of +5mm/yr and LIA termination date of 1900 CE. d) As part c except grey*
*shaded area corresponds to RSL trends from Figure 6b, demonstrating a statistically equivalent fit to*
*the data. For illustrative purposes, the red line denotes a modelled RSL scenario with a budget residual*
*rate of +2.5mm/yr and LIA termination date of 1800 CE; the blue line a modelled RSL scenario with a*
*budget residual rate of +5mm/yr and LIA termination date of 1900 CE.*

*4.2   Reliability of saltmarsh RSL data*
Saltmarshes and their microfossil communities are widely used in temperate locations and previously
in west and south Greenland to reconstruct recent RSL changes with high precision (e.g. Kemp et al.,
2009, 2017; Long et al., 2012, 2010; Woodroffe and Long, 2009).  At Dronning Marie Dal, the first
half of the RSL record (1725-1880 CE) is harder to interpret because early, rapid RSL rise may indicate
either a local LIA loading signal or a non RSL factor (e.g. sediment supply changes) as the marsh
became established.  What we can say with certainty is that RSL began to fall at or soon after 1880 CE,
suggesting additional contributors to RSL or changes in the dominance of existing contributors caused
this change in the sign and rate of RSL.  We are also confident of the total amount of RSL fall between
1880 CE and present, which is less than predicted by any permutation of the sea-level budget modelling
(Figure 5).  We acknowledge however that these reconstructions come from a single sediment core and
although the stratigraphy appeared consistent across the marsh during fieldwork it would be ideal to
replicate these results within another core from the same marsh and also from other marshes close to
the ice sheet margin in this region in the future.

There is no indication of hiatuses within the marsh sediment and based on surveys of modern marshes
here and elsewhere in Greenland the elevation range of the key diatom species *Pinnularia intermedia*
used in the palaeo-marsh surface elevation calculations is robust.  A RSL fall of ~0.6-0.9 m since 1900
CE as predicted by the sea-level budget modelling, would have lifted what was a high marsh
environment at the start of the period (indicated by the taxa at ~5 cm, Figure 3B) out of the intertidal
and into the adjacent freshwater zone where diatoms are not preserved due to extreme aridity.  The



continuous preservation of intertidal diatoms through the sediment sequence to the surface where
modern saltmarsh plants were growing during sampling (Figure 2) rules out this possibility.  Even the
smaller amount of RSL fall (~0.2 m) since 1900 CE predicted by an earlier LIA termination date (1800
CE) and 30% smaller GrIS contribution (Figure 6) is unlikely because the diatoms suggest a mid-high
marsh environment at 1900 CE and the core top elevation is within the high marsh zone, a vertical
distance based on analysis of modern diatoms at Dronning Marie Dal of ~0.1 m, which is half of the
predicted RSL fall (~0.2 m).  Greenland saltmarshes accrete very slowly and only record sustained RSL
changes over decades, and therefore short-timescale variability in contributors (e.g. due to decadal
temperature fluctuations in the 20th Century) is not distinguishable in the saltmarsh data.  However, the
total amount of RSL fall and the timing of the change from RSL rise/stability to RSL fall is robustly
reconstructed and we are confident that this provides an important test of Greenland RSL modelling.

*4.3 Limitations of RSL modelling*
Regional sea level budgets deviate significantly from the global budget, are challenging to compute and
have been deemed part of the 'Regional Sea-Level Change and Coastal Impacts' Grand Challenge by
the World Climate Research Programme (WCRP, 2022). Of the different items in the sea-level budget
for Dronning Marie Dal, the large uncertainty in the steric contribution could potentially be the source
of additional sea-level rise which would help decrease the 'budget residual' identified in Figure 6.  The
data in Table 4 do not fully capture the range of uncertainty in the steric component of sea level.  These
uncertainties arise from poor to non-existent capture of the dynamics of coastal regions, namely the
propagation of the change in steric height of the open ocean to the fjord location and the lack of
observations to constrain model output in the early 20th Century.

The field site is located at the head of the 50 km long marine fjord Søndre Skoldungesund and therefore
the steric contribution may be different to that calculated from the open ocean estimates within 300 km
of Dronning Marie Dal averaged in this study.  A multibeam study of the fjord by Kjeldsen et al. (2017)
shows the fjord is between 1.1-3.1 km wide, up to 800 m deep in the outer part, with a shallow (77m
deep) sill at mid-fjord and shallow depths inside the sill.  The fjord water is cold to the base along its
length, with no apparent intrusion of warmer Atlantic water from the shelf edge.  The mixed predictions




of steric height changes from the different models suggest that this region is poorly constrained within
global steric datasets (Table 4).  Given the lack of intrusion of warm Atlantic water into the fjord today
it is unlikely that there has been a more positive contribution of steric height from 20[th] Century warming.
However, with significant mass loss from the Greenland Ice Sheet since the LIA and an influx of cold
yet low-salinity meltwater into the fjord it is possible that the local halosteric component is
underestimated.

A second issue with the steric height calculation is the potential for the CMIP6 models to misrepresent
changes in the dynamic height of the ocean caused by shifts in the location of ocean currents, such as
the East Greenland Current (EGC) over time.  A recent study of North Atlantic dynamic sea level and
its response to GrIS meltwater and temperature increase indicates general Atlantic Meridional
Overturning Circulation decline and increase in sea-surface height with increased GrIS melting, but the
response of the cold EGC is complex and in southeast Greenland the effect of warming and increased
meltwater on sea-surface height is minimal (Saenko et al., 2017).  Given that Kjeldsen et al. (2017)
suggest the EGC does not currently penetrate into the Søndre Skoldungesund fjord the impact of any
dynamical changes in the EGC since the LIA are likely to be minor.

A third possible source of uncertainty in the sea-level budget is the application of the sea level code
used to calculate GIA, specifically the spectral resolution with which the algorithm predicts the sea
level response to loading increments. The mass balance history from Kjeldsen et al. (2015) is presented
on a 1x1 km spatial grid, but the sea level code utilises a spectral harmonic truncation of 256. The
effects on predicted RSL of the reduction in resolution has been demonstrated previously with near-
field relative sea level being more affected by harmonic truncation than far field sites (Spada and Melini,
2019). A move towards a higher degree spherical harmonic truncation (>1024) would be necessary to
faithfully reproduce sea level fingerprint histories associated with small outlet glaciers and should be
considered in the future (Adhikari et al., 2015).

Despite the limitations outlined above, this study presents a first test of a post-LIA sea-level budget in
the nearfield location of southeast Greenland.  There is clear and unexplained difference between the





RSL history recorded by salt marsh sediments (a small RSL fall since the end of the LIA) and the RSL
budget which suggests significant RSL fall during this period.  The sensitivity tests show that the budget
can fit the salt marsh RSL data if the amount of mass loss from the GrIS and peripheral glaciers is less,
and it took place over a longer period (Figure 6d), but even so a +2.5 mm/yr unexplained 'budget
residual' remains.  RSL reconstructions from salt marshes in southwest Greenland (Long et al., 2012,
2010; Woodroffe and Long, 2010, 2009) also suggest that the dominant signal in southern Greenland is
RSL rise into the 20[th] Century, which correlates with the long term (pre ~1880 CE) trend of RSL rise at
Dronning Marie Dal.

**5. Conclusions**
Saltmarsh sediments collected at the mouth of Dronning Marie Dal, close to the GrIS margin in
southeast Greenland, record RSL changes over the past c. 300 years in changing sediment and diatom
stratigraphy.   These RSL changes record a combination of processes that are dominated by
local/regional changes in GrIS mass balance during this critical period that spans the maximum of the
LIA and 20th Century warming.

In the early part of the record (1725-1762 CE) the rate of RSL rise is higher than reconstructed from
the closest isolation basin at Timmiarmiut, but between 1762-1880 CE the rate decreases to within the
error range of the isolation basin RSL rate.  This trend is likely due to changes in the local LIA ice load
over this time period combined with ongoing millennial-scale GIA, or other local processes as the salt
marsh is established.  Other recent estimates of centennial-scale GIA (Khan et al., 2016; van Dam et
al., 2017) suggest that RSL should have been falling over the past few hundred years at Dronning Marie
Dal.  The isolation basin and salt marsh data instead suggest that RSL was rising or close to stable from
c. 1100 CE until c. 1880 CE.  RSL begins to slowly fall around 1880 CE and the rate then accelerates
from the 1990s, with a total amount of RSL fall of 0.08 ± 0.1 m since 1880 CE.







Modelled RSL, which takes into account contributions from post-LIA GrIS GIA, ongoing deglacial
GIA, the global non-ice sheet glacial fingerprint, the contribution from thermosteric effects, an estimate
of the Antarctic fingerprint and the contribution from terrestrial water storage, over-predicts the amount
of RSL fall since the end of the LIA by at least 0.5 m. The GIA signal caused by post-LIA GrIS mass
loss is by far the largest contributor, and error in its calculation has the largest potential to impact RSL
predictions at Dronning Marie Dal. We cannot reconcile the modelled contributions and the saltmarsh
observations, even when moving the termination of the LIA to 1800 CE, and reducing the post-LIA
Greenland mass loss signal by 30%. A 'budget residual' of ~+2.5 mm/yr since the end of the LIA
remains unexplained. Explaining the difference between salt marsh RSL data and the modelled RSL
budget since the end of the LIA should be a key future research objective through reducing uncertainty
on each component to the sea-level budget, collecting more empirical data on the recent history of the
GrIS and by replicating the salt marsh RSL record presented here elsewhere in this and other regions of
Greenland.

*Author Contribution*
SAW, LMW, AJL and KKK designed the study, SAW, KKK and KHK undertook fieldwork, NLMB
undertook the laboratory analysis, and SAW and LMW prepared the manuscript with contributions
from all co-authors.

The authors declare that they have no conflict of interest.

*Acknowledgements*
We acknowledge funding from Danish Agency for Science, Technology and Innovation, 'Greenland
ice sheet over the past millennium' (PI Kurt Kjær) and the assistance of the captain and crew onboard
SS ACTIV for their help collecting the data during the field campaign to southeast Greenland. Barlow's
postdoctoral position to undertake this work was funded by Durham University Department of
Geography. We thank the laboratory technicians within Durham Geography for their support with
sample preparation. The authors acknowledge the International Union for Quaternary Sciences
(INQUA) Coastal and Marine Processes (CMP) Commission and PALSEA, a working group of INQUA



and Past Global Changes (PAGES), which in turn receives support from the Swiss Academy of Sciences
and the Chinese Academy of Sciences.

Adhikari, S., Ivins, E. R., and Larour, E.: ISSM-SESAW v1.0: mesh-based computation of
gravitationally consistent sea level and geodetic signatures caused by cryosphere and
climate driven mass change, Climate and Earth System Modeling,
https://doi.org/10.5194/gmdd-8-9769-2015, 2015.
Adhikari, S., Caron, L., Steinberger, B., Reager, J. T., Kjeldsen, K. K., Marzeion, B., Larour,
E., and Ivins, E. R.: What drives 20th century polar motion?, Earth Planet. Sci. Lett., 502,
126–132, https://doi.org/10.1016/j.epsl.2018.08.059, 2018.
Adhikari, S., Milne, G. A., Caron, L., Khan, S. A., Kjeldsen, K. K., Nilsson, J., Larour, E., and
Ivins, E. R.: Decadal to Centennial Timescale Mantle Viscosity Inferred from Modern Crustal
Uplift Rates in Greenland, Geophys. Res. Lett., n/a, e2021GL094040,
https://doi.org/10.1029/2021GL094040, 2021.
Allen, J. R. L.: Morphodynamics of Holocene salt marshes: a review sketch from the Atlantic
and Southern North Sea coasts of Europe, Quat. Sci. Rev., 19, 1155–1231,
https://doi.org/Doi 10.1016/S0277-3791(99)00034-7, 2000.
Bamber, J. and Riva, R.: The sea level fingerprint of recent ice mass fluxes, The
Cryosphere, 4, 621–627, https://doi.org/10.5194/tc-4-621-2010, 2010.
Barlow, N. L. M., Shennan, I., Long, A. J., Gehrels, W. R., Saher, M. H., Woodroffe, S. A.,
and Hillier, C.: Salt marshes as late Holocene tide gauges, Glob. Planet. Change, 106, 90–
110, https://doi.org/10.1016/j.gloplacha.2013.03.003, 2013.
Bevis, M., Wahr, J., Khan, S. A., Madsen, F. B., Brown, A., Willis, M., Kendrick, E., Knudsen,
P., Box, J. E., van Dam, T., Caccamise, D. J., Johns, B., Nylen, T., Abbott, R., White, S.,
Miner, J., Forsberg, R., Zhou, H., Wang, J., Wilson, T., Bromwich, D., and Francis, O.:
Bedrock displacements in Greenland manifest ice mass variations, climate cycles and
climate change, Proc. Natl. Acad. Sci. U. S. A., 109, 11944–11948, https://doi.org/DOI
10.1073/pnas.1204664109, 2012.
Bevis, M., Harig, C., Khan, S. A., Brown, A., Simons, F. J., Willis, M., Fettweis, X., Broeke,
M. R. van den, Madsen, F. B., Kendrick, E., Caccamise, D. J., Dam, T. van, Knudsen, P.,
and Nylen, T.: Accelerating changes in ice mass within Greenland, and the ice sheet's
sensitivity to atmospheric forcing, Proc. Natl. Acad. Sci., 116, 1934–1939,
https://doi.org/10.1073/pnas.1806562116, 2019.
Bindler, R., Renberg, I., Appleby, P. G., Anderson, N. J., and Rose, N. L.: Mercury
Accumulation Rates and Spatial Patterns in Lake Sediments from West Greenland: A Coast
to Ice Margin Transect, Environ. Sci. Technol., 35, 1736–1741,
https://doi.org/10.1021/es0002868, 2001.
Bjork, A. A., Kjaer, K. H., Korsgaard, N. J., Khan, S. A., Kjeldsen, K. K., Andresen, C. S.,
Box, J. E., Larsen, N. K., and Funder, S.: An aerial view of 80 years of climate-related glacier
fluctuations in southeast Greenland, Nat. Geosci., 5, 427–432,
https://doi.org/10.1038/Ngeo1481, 2012.
Briner, J. P., Young, N. E., Thomas, E. K., Stewart, H. A. M., Losee, S., and Truex, S.: Varve
and radiocarbon dating support the rapid advance of Jakobshavn Isbræ during the Little Ice
Age, Quat. Sci. Rev., 30, 2476–2486, https://doi.org/10.1016/j.quascirev.2011.05.017, 2011.



Briner, J. P., Cuzzone, J. K., Badgeley, J. A., Young, N. E., Steig, E. J., Morlighem, M.,
Schlegel, N.-J., Hakim, G. J., Schaefer, J. M., Johnson, J. V., Lesnek, A. J., Thomas, E. K.,
Allan, E., Bennike, O., Cluett, A. A., Csatho, B., de Vernal, A., Downs, J., Larour, E., and
Nowicki, S.: Rate of mass loss from the Greenland Ice Sheet will exceed Holocene values
this century, Nature, 586, 70–74, https://doi.org/10.1038/s41586-020-2742-6, 2020.
van den Broeke, M., Bamber, J., Ettema, J., Rignot, E., Schrama, E., van de Berg, W., van
Meijgaard, E., Velicogna, I., and Wouters, B.: Partitioning Recent Greenland Mass Loss,
Science, 326, 984–986, https://doi.org/10.1126/science.1178176, 2009.
Bronk Ramsey, C.: Bayesian analysis of radiocarbon dates, Radiocarbon, 51, 337–360,
755   2009.

Chao, B. F., Wu, Y. H., and Li, Y. S.: Impact of Artificial Reservoir Water Impoundment on
Global Sea Level, Science, 320, 212–214, https://doi.org/10.1126/science.1154580, 2008.
Chen, G., Zhang, S., Liang, S., and Zhu, J.: Elevation and Volume Changes in Greenland
Ice Sheet From 2010 to 2019 Derived From Altimetry Data, Front. Earth Sci., 9, 674983,
https://doi.org/10.3389/feart.2021.674983, 2021.
Chylek, P., Dubey, M. K., and Lesins, G.: Greenland warming of 1920-1930 and 1995-2005,
Geophys. Res. Lett., 33, https://doi.org/Artn L11707 10.1029/2006gl026510, 2006.
van Dam, T., Francis, O., Wahr, J., Khan, S. A., Bevis, M., and van den Broeke, M. R.: Using
GPS and absolute gravity observations to separate the effects of present-day and
Pleistocene ice-mass changes in South East Greenland, Earth Planet. Sci. Lett., 459, 127–
135, https://doi.org/10.1016/j.epsl.2016.11.014, 2017.
Döll, P., Müller Schmied, H., Schuh, C., Portmann, F. T., and Eicker, A.: Global-scale
assessment of groundwater depletion and related groundwater abstractions: Combining
hydrological modeling with information from well observations and GRACE satellites, Water
Resour. Res., 50, 5698–5720, https://doi.org/10.1002/2014WR015595, 2014.
Dyke, L. M., Hughes, A. L. C., Murray, T., Hiemstra, J. F., Andresen, C. S., and Rodés, Á.:
Evidence for the asynchronous retreat of large outlet glaciers in southeast Greenland at the
end of the last glaciation, Quat. Sci. Rev., 99, 244–259,
https://doi.org/10.1016/j.quascirev.2014.06.001, 2014.
Dyke, L. M., Hughes, A. L., Andresen, C. S., Murray, T., Hiemstra, J. F., Bjørk, A. A., and
Rodés, Á.: The deglaciation of coastal areas of southeast Greenland, The Holocene, 28,
1535–1544, https://doi.org/10.1177/0959683618777067, 2018.
Dziewonski, A. M. and Anderson, D. L.: Preliminary reference Earth model, Phys. Earth
Planet. Inter., 25, 297–356, https://doi.org/10.1016/0031-9201(81)90046-7, 1981.
Eyring, V., Bony, S., Meehl, G. A., Senior, C. A., Stevens, B., Stouffer, R. J., and Taylor, K.
E.: Overview of the Coupled Model Intercomparison Project Phase 6 (CMIP6) experimental
design and organization, Geosci. Model Dev., 9, 1937–1958, https://doi.org/10.5194/gmd-9-
783   1937-2016, 2016.

Farrell, W. E. and Clark, J. A.: On Postglacial Sea Level, Geophys. J. R. Astron. Soc., 46,
647–667, https://doi.org/10.1111/j.1365-246X.1976.tb01252.x, 1976.
Frederikse, T., Landerer, F., Caron, L., Adhikari, S., Parkes, D., Humphrey, V. W.,
Dangendorf, S., Hogarth, P., Zanna, L., Cheng, L., and Wu, Y.-H.: The causes of sea-level
rise since 1900, Nature, 584, 393–397, https://doi.org/10.1038/s41586-020-2591-3, 2020.





Funder, S. and Hansen, L.: The Greenland ice sheet - a model for its culmination and decay
during and after the last glacial maximum, Bull. Geol. Soc. Den., 42, 137–152, 1996.
Funder, S., Kjeldsen, K. K., Kjaer, K. H., and O Cofaigh, C.: The Greenland Ice Sheet during
the last 300,000 years: a review, Dev. Quat. Sci., 15, 699-713. doi: 10.1016/B978-0-444-
793   53447-7.00050-7, 2011.

Hughes, A., Rainsley, E., Murray, T., Fogwill, C., Schnabel, C., and Xu, S.: Rapid response
of Helheim Glacier, southeast Greenland, to early Holocene climate warming, Geology, 40,
427–430, https://doi.org/10.1130/G32730.1, 2012.
Humphrey, V. and Gudmundsson, L.: GRACE-REC: a reconstruction of climate-driven water
storage changes over the last century, Earth Syst. Sci. Data, 11, 1153–1170,
https://doi.org/10.5194/essd-11-1153-2019, 2019.
Ivchenko, V. O., Danilov, S., Sidorenko, D., Schröter, J., Wenzel, M., and Aleynik, D. L.:
Steric height variability in the Northern Atlantic on seasonal and interannual scales, J.
Geophys. Res., 113, C11007, https://doi.org/10.1029/2008JC004836, 2008.
Kemp, A., Horton, B., Culver, S., Corbett, D., van de Plassche, O., Gehrels, W., Douglas, B.,
and Parnell, A.: Timing and magnitude of recent accelerated sea-level rise (North Carolina,
United States), Geology, 37, 1035–1038, https://doi.org/10.1130/G30352A.1, 2009.
Kemp, A. C., Kegel, J. J., Culver, S. J., Barber, D. C., Mallinson, D. J., Leorri, E., Bernhardt,
C. E., Cahill, N., Riggs, S. R., Woodson, A. L., Mulligan, R. P., and Horton, B. P.: Extended
late Holocene relative sea-level histories for North Carolina, USA, Quat. Sci. Rev., 160, 13–
30, https://doi.org/10.1016/j.quascirev.2017.01.012, 2017.
Kendall, R. A., Mitrovica, J. X., and Milne, G. A.: On post-glacial sea level - II. Numerical
formulation and comparative results on spherically symmetric models, Geophys. J. Int., 161,
679–706, https://doi.org/10.1111/j.1365-246X.2005.02553.x, 2005.
Khan, S. A., Aschwanden, A., Bjork, A. A., Wahr, J., Kjeldsen, K. K., and Kjaer, K. H.:
Greenland ice sheet mass balance: a review, Rep. Prog. Phys., 78, 26,
https://doi.org/10.1088/0034-4885/78/4/046801, 2015.
Khan, S. A., Sasgen, I., Bevis, M., van Dam, T., Bamber, J. L., Wahr, J., Willis, M., Kjaer, K.
H., Wouters, B., Helm, V., Csatho, B., Fleming, K., Bjork, A. A., Aschwanden, A., Knudsen,
P., and Munneke, P. K.: Geodetic measurements reveal similarities between post-Last
Glacial Maximum and present-day mass loss from the Greenland ice sheet, Sci. Adv., 2,
https://doi.org/ARTN e1600931 10.1126/sciadv.1600931, 2016.
Khan, S. A., Bjørk, A. A., Bamber, J. L., Morlighem, M., Bevis, M., Kjær, K. H., Mouginot, J.,
Løkkegaard, A., Holland, D. M., Aschwanden, A., Zhang, B., Helm, V., Korsgaard, N. J.,
Colgan, W., Larsen, N. K., Liu, L., Hansen, K., Barletta, V., Dahl-Jensen, T. S.,
Søndergaard, A. S., Csatho, B. M., Sasgen, I., Box, J., and Schenk, T.: Centennial response
of Greenland's three largest outlet glaciers, Nat. Commun., 11, 5718,
https://doi.org/10.1038/s41467-020-19580-5, 2020.
Kjær, K. H., Bjørk, A. A., Kjeldsen, K. K., Hansen, E. S., Andresen, C. S., Siggaard-
Andersen, M.-L., Khan, S. A., Søndergaard, A. S., Colgan, W., Schomacker, A., Woodroffe,
S., Funder, S., Rouillard, A., Jensen, J. F., and Larsen, N. K.: Glacier response to the Little
Ice Age during the Neoglacial cooling in Greenland, Earth-Sci. Rev., 227, 103984,
https://doi.org/10.1016/j.earscirev.2022.103984, 2022.
Kjeldsen, K., Korsgaard, N., Bjork, A., Khan, S., Box, J., Funder, S., Larsen, N., Bamber, J.,
Colgan, W., van den Broeke, M., Siggaard-Andersen, M., Nuth, C., Schomacker, A.,
Andresen, C., Willerslev, E., and Kjaer, K.: Spatial and temporal distribution of mass loss





from the Greenland Ice Sheet since AD 1900, Nature, 528, 396–400,
https://doi.org/10.1038/nature16183, 2015.

Kjeldsen, K. K., Weinrebe, R. W., Bendtsen, J., Bjørk, A. A., and Kjær, K. H.: Multibeam
bathymetry and CTD measurements in two fjord systems in southeastern Greenland, Earth
Syst. Sci. Data, 9, 589–600, https://doi.org/10.5194/essd-9-589-2017, 2017.

Lecavalier, B., Milne, G. A., Simpson, M. J. R., Wake, L. M., Huybrechts, P., Tarasov, L.,
Kjeldsen, K. K., Funder, S. V., Long, A. J., Woodroffe, S. A., Dyke, A., and Larsen, N. K.: A
model of Greenland ice sheet deglaciation based on observations of relative sea-level and
ice extent, Quat. Sci. Rev., in press, 2014.

Lepping, O. and Daniëls, F. J. A.: Phytosociology of Beach and Salt Marsh Vegetation in
Northern West Greenland, Polarforschung, 76, 95–108, 2007.

Levy, L. B., Larsen, N. K., Knudsen, M. F., Egholm, D. L., Bjørk, A. A., Kjeldsen, K. K., Kelly,
M. A., Howley, J. A., Olsen, J., Tikhomirov, D., Zimmerman, S. R. H., and Kjær, K. H.: Multi-
phased deglaciation of south and southeast Greenland controlled by climate and
topographic setting, Quat. Sci. Rev., 242, 106454,
https://doi.org/10.1016/j.quascirev.2020.106454, 2020.

Lindeberg, C., Bindler, R., Renberg, I., Emteryd, O., Karlsson, E., and Anderson, N. J.:
Natural Fluctuations of Mercury and Lead in Greenland Lake Sediments, Environ. Sci.
Technol., 40, 90–95, https://doi.org/10.1021/es051223y, 2006.

Long, A. J., Woodroffe, S. A., Milne, G. A., Bryant, C. L., and Wake, L. M.: Relative sea-level
change in West Greenland during the last millennium, Quat. Sci. Rev., 29, 367–383, 2010.

Long, A. J., Woodroffe, S. A., Milne, G. A., Bryant, C. L., Simpson, M. J. R., and Wake, L.
M.: Relative sea-level change in Greenland during the last 700 yrs and ice sheet response to
the Little Ice Age, Earth Planet. Sci. Lett., 315, 76–85, https://doi.org/DOI
10.1016/j.epsl.2011.06.027, 2012.

Marzeion, B., Jarosch, A. H., and Hofer, M.: Past and future sea-level change from the
surface mass balance of glaciers, The Cryosphere, 6, 1295–1322, https://doi.org/10.5194/tc-
6-1295-2012, 2012.

Marzeion, B., Leclercq, P. W., Cogley, J. G., and Jarosch, A. H.: Brief Communication:
Global reconstructions of glacier mass change during the 20th century are consistent, The
Cryosphere, 9, 2399–2404, https://doi.org/10.5194/tc-9-2399-2015, 2015.

McDougall, T. J. and Barker, P. M.: Getting started with TEOS-10 and the Gibbs Seawater
(GSW) Oceanographic Toolbox., 2011.

Meredith, M., Sommerkorn, M., Cassotta, S., Derksen, C., Ekaykin, A., Hollowed, A.,
Kofinas, G., Mackintosh, A., Melbourne-Thomas, J., Muelbert, M. M. C., Ottersen, G.,
Pritchard, H., and Schuur, E. A. G.: Polar Regions, in: IPCC Special Report on the Ocean
and Cryosphere in a Changing Climate, Cambridge University Press, 203–320, 2019.

Mitrovica, J. X. and Milne, G. A.: On post-glacial sea level: I. General theory, Geophys. J.
Int., 154, 253–267, https://doi.org/DOI 10.1046/j.1365-246X.2003.01942.x, 2003.

Mitrovica, J. X., Tamisiea, M. E., Davis, J. L., and Milne, G. A.: Recent mass balance of
polar ice sheets inferred from patterns of global sea-level change, Nature, 409, 1026–1029,
https://doi.org/10.1038/35059054, 2001.

Moon, T., Joughin, I., Smith, B., and Howat, I.: 21st-Century Evolution of Greenland Outlet
Glacier Velocities, Science, 336, 576–578, https://doi.org/10.1126/science.1219985, 2012.



Morlighem, M., Williams, C. N., Rignot, E., An, L., Arndt, J. E., Bamber, J. L., Catania, G.,
Chauché, N., Dowdeswell, J. A., Dorschel, B., Fenty, I., Hogan, K., Howat, I., Hubbard, A.,
Jakobsson, M., Jordan, T. M., Kjeldsen, K. K., Millan, R., Mayer, L., Mouginot, J., Noël, B. P.
Y., O'Cofaigh, C., Palmer, S., Rysgaard, S., Seroussi, H., Siegert, M. J., Slabon, P., Straneo,
F., van den Broeke, M. R., Weinrebe, W., Wood, M., and Zinglersen, K. B.: BedMachine v3:
Complete Bed Topography and Ocean Bathymetry Mapping of Greenland From Multibeam
Echo Sounding Combined With Mass Conservation, Geophys. Res. Lett., 44, 11,051-
11,061, https://doi.org/10.1002/2017GL074954, 2017.
Pérez-Rodríguez, M., Silva-Sánchez, N., Kylander, M. E., Bindler, R., Mighall, T. M.,
Schofield, J. E., Edwards, K. J., and Martínez Cortizas, A.: Industrial-era lead and mercury
contamination in southern Greenland implicates North American sources, Sci. Total
Environ., 613–614, 919–930, https://doi.org/10.1016/j.scitotenv.2017.09.041, 2018.
Pritchard, H., Arthern, R., Vaughan, D., and Edwards, L.: Extensive dynamic thinning on the
margins of the Greenland and Antarctic ice sheets, Nature, 461, 971–975,
https://doi.org/10.1038/nature08471, 2009.
Ramsey, C. B. and Lee, S.: Recent and Planned Developments of the Program OxCal,
Radiocarbon, 55, 720–730, https://doi.org/10.1017/S0033822200057878, 2013.
Reimer, P. J., Austin, W. E. N., Bard, E., Bayliss, A., Blackwell, P. G., Ramsey, C. B., Butzin,
M., Cheng, H., Edwards, R. L., Friedrich, M., Grootes, P. M., Guilderson, T. P., Hajdas, I.,
Heaton, T. J., Hogg, A. G., Hughen, K. A., Kromer, B., Manning, S. W., Muscheler, R.,
Palmer, J. G., Pearson, C., Plicht, J. van der, Reimer, R. W., Richards, D. A., Scott, E. M.,
Southon, J. R., Turney, C. S. M., Wacker, L., Adolphi, F., Büntgen, U., Capano, M., Fahrni,
S. M., Fogtmann-Schulz, A., Friedrich, R., Köhler, P., Kudsk, S., Miyake, F., Olsen, J.,
Reinig, F., Sakamoto, M., Sookdeo, A., and Talamo, S.: The IntCal20 Northern Hemisphere
Radiocarbon Age Calibration Curve (0–55 cal kBP), Radiocarbon, 62, 725–757,
https://doi.org/10.1017/RDC.2020.41, 2020.
Richter, A., Rysgaard, S., Dietrich, R., Mortensen, J., and Petersen, D.: Coastal tides in
West Greenland derived from tide gauge records, Ocean Dyn., 61, 39–49,
https://doi.org/10.1007/s10236-010-0341-z, 2011.
Saenko, O. A., Yang, D., and Myers, P. G.: Response of the North Atlantic dynamic sea
level and circulation to Greenland meltwater and climate change in an eddy-permitting ocean
model, Clim. Dyn., 49, 2895–2910, https://doi.org/10.1007/s00382-016-3495-7, 2017.
Shotyk, W., Goodsite, M. E., Roos-Barraclough, F., Frei, R., Heinemeier, J., Asmund, G.,
Lohse, C., and Hansen, T. S.: Anthropogenic contributions to atmospheric Hg, Pb and As
accumulation recorded by peat cores from southern Greenland and Denmark dated using
the 14C "bomb pulse curve," Geochim. Cosmochim. Acta, 67, 3991–4011,
https://doi.org/10.1016/S0016-7037(03)00409-5, 2003.
Spada, G. and Melini, D.: SELEN4; (SELEN version 4.0): a Fortran program for solving the
gravitationally and topographically self-consistent sea-level equation in glacial isostatic
adjustment modeling, Geosci. Model Dev., 12, 5055–5075, https://doi.org/10.5194/gmd-12-
919 5055-2019, 2019.

The IMBIE Team: Mass balance of the Greenland Ice Sheet from 1992 to 2018, Nature, 579,
233–239, https://doi.org/10.1038/s41586-019-1855-2, 2020.
Vogt, T.: Late-Quaternary Oscillations of Level in Southeast Greenland., Oslo: Dybwad
1933., 44 pp., 1933.





Wada, Y., Lo, M.-H., Yeh, P. J.-F., Reager, J. T., Famiglietti, J. S., Wu, R.-J., and Tseng, Y.-
H.: Fate of water pumped from underground and contributions to sea-level rise, Nat. Clim.
Change, 6, 777–780, https://doi.org/10.1038/nclimate3001, 2016.
Wangner, D. J., Sicre, M., Kjeldsen, K. K., Jaeger, J. M., Bjørk, A. A., Vermassen, F., Sha,
L., Kjær, K. H., Klein, V., and Andresen, C. S.: Sea Surface Temperature Variability on the
SE-Greenland Shelf (1796–2013 CE) and Its Influence on Thrym Glacier in Nørre
Skjoldungesund, Paleoceanogr. Paleoclimatology, 35,
https://doi.org/10.1029/2019PA003692, 2020.
Wood, K. R. and Overland, J. E.: Early 20th century Arctic warming in retrospect, Int. J.
Climatol., 30, 1269–1279, https://doi.org/10.1002/joc.1973, 2010.
Woodroffe, S. A. and Long, A. J.: Salt marshes as archives of recent relative sea-level
change in West Greenland, Quat. Sci. Rev., 28, 1750–1761, 2009.
Woodroffe, S. A. and Long, A. J.: Reconstructing recent relative sea-level changes in West
Greenland: local diatom-based transfer functions are superior to regional models, Quat. Int.,
221, 91–103, 2010.
Zheng, J.: Archives of total mercury reconstructed with ice and snow from Greenland and
the Canadian High Arctic, Sci. Total Environ., 509–510, 133–144,
https://doi.org/10.1016/j.scitotenv.2014.05.078, 2015.

