# Peer review of "Missing sea-level rise in southeast Greenland during and since"

_EGUsphere, 2022_

## Author Response (AR1)

**Robin Edwards: General Comments**

This paper compares a 300 year record of relative sea-level (RSL) change in southeast Greenland with a sea level budget developed from a combination of observations and modelling. The authors conclude that there is a significant mismatch between the amount/rate of RSL change inferred from the proxy-based reconstruction and the sum of modelled contributions, indicating errors in one or both of these datasets/approaches. Whilst the authors conduct some sensitivity analysis in an attempt to identify likely sources of error, they ultimately conclude that the source of the misfit remains unexplained, with the challenge for future work to reduce the 'budget residual' of +2.5 mm/yr since the end of the Little Ice Age (LIA).

The paper addresses a scientific question of relevance to this journal, presents new RSL data from a poorly studied location / period, and identifies a significant limitation in current understanding that will require new data and analysis to resolve. I cannot comment on the details of the modelling components of the study, but the methods used to produce the RSL reconstruction are sound and reasonably well-established. The substantive conclusion of a mismatch between modelled RSL budget and the field evidence for RSL change appears robust and I recommend this paper for publication. I have several questions relating to the more detailed inferences of elevated rates of RSL change in the early and latest portions of the record (see specific comments) and some suggested modifications to tables / figures. Consequently I recommend publication subject to minor revision.

**We thank Dr Edwards for his helpful and constructive comments on the manuscript and we respond to these below in red.**

**Specific Comments**

**Calculation of palaeomarsh-surface elevation (PMSE)**

The authors use a visual assessment of indicator species to essentially delimit vertical assemblage zones that are used to infer a particular PMSE with associated uncertainty (as in Table S2). The basic approach is sound but I have a few questions about its application to the particular dataset that the authors may be able to clarify.

The main elements of the visual assessment criteria appear well suited to identifying which side of the high marsh / upland boundary and low marsh / high marsh boundary a sample sits, but I have a question regarding how the finer-grained differentiations were established (e.g. the difference between PI 5%, PI 6%, PI 10%) which are based on very small percentage differences. Can these 'zones' be clearly identified on Figure 3A (if not, perhaps they are not reliable)? What was the count size of the diatom samples in both the surface and fossil datasets and is this sufficient to accurately identify a 1% difference? The reason for raising this question is that the current PMSE zones create the very smooth / progressive change in PMSE noted at the top of the core which contributes to the inferred 20th century rate change discussed in the text (also see comment on top of the core below).

The visual assessment method used in the paper to infer palaeo-marsh surface elevations is a mixed approach which uses zones of the marsh inhabited by key diatom species to infer palaeo-marsh surface elevation, and additional information provided by the lithology and nature of change between adjacent diatom samples.

Using zones alone to reconstruct RSL introduces artificial jumps in the RSL record when moving from a sample reconstructed from within one zone to the next sample in a different zone. To create an RSL reconstruction with no artificial jumps within it we use a smoothing function which allows the PMSE to change within each zone, noting the progressive way that the key diatom taxa change up core. For instance the progressive rise in Pinnularia intermedia in the top 4 cm suggests smoothly falling RSL during this period. We therefore modify the PMSE results within this zone to allow for the progressive change seen in the diatoms. This is

backed up by the LOI data which suggests a progressive rise in organic content in the top 4 cm indicative of rising PMSE.

We have added text to the methods and results sections, and table 2 in the Supplementary Information to make it clearer how the PMSE values have been calculated which explains that the method uses understanding gained from the modern distribution of diatoms across the marsh, but also the nature of change between individual samples both in terms of diatom distribution and lithology (e.g. LOI values).

To answer the specific questions posed above in relation to this issue, table 2 in Supp Info currently (for the top few samples) lists the % of Pinnularia intermedia in the sample, not strictly the assemblage zone used to create the PMSE. We have edited the information in this table as we agree that it is misleading and suggests precision that is not possible to resolve.

The PMSE between 5 cm and 8 cm depth (corresponding to the 19th century, essentially stable RSL interval) is 1.65 m with a 15 cm error bar on either side. I'm not entirely clear where this comes from and wonder whether the authors could be slightly less conservative here? For example, at and below the low marsh / high marsh boundary, the proportions of Achnanthesspecies and Navicula salinarum rise to relative abundances similar to or greater than those used for the favoured indicator species PI and NC. According to Fig 3 B neither of these taxa are present in the core. Unless there are known issues of selective preservation which means these taxa a likely to have been preferentially removed, their absence would seem to argue for accumulation above the low marsh/high marsh boundary. This effectively precludes the 1.5 - 1.65m PMSE range associated with the lower error bars.

We have reconsidered the species zones used to calculate the PMSE values in the mid part of the core and agree that we have been too conservative in assigning errors as additional information is available because of the absence of Achnanthes sp. and Navicula salinarum in these samples. We have amended the criteria used to calculate the PMSE values for these samples in table 2 in the supplementary information, amended figures where the RSL data is presented and checked the text to ensure it reflects these new RSL values. These changes do not make material difference to the take-home messages of the paper.

The potential significance of this is that, if correct, it would elevate the mid-point of the [PI below 5%, NC above 5%] zone, bringing it close to 1.75 - 1.8 m. In effect, the PMSE is now the mid-point of a high marsh zone delimited by the two grey shaded boundaries on Fig 3A (assuming that there are symmetrical error terms). If this is a correct interpretation of the data, this greatly reduces the apparent RSL rise at the bottom of the core which is purely driven by the magnitude of the change in PMSE (ie elevating the PMSE, reduces the inferred RSL rise).

Re-evaluating PMSE values for the central part of the core does reduce a little the magnitude of RSL change across the lower half of the core. However the lower-most samples are still reconstructed to be from the very highest part of the marsh and therefore we still reconstruct the largest amount of RSL rise in this section. We have made the proposed changes to the figures and text in the methods, results and discussion.

Accuracy of the accumulation history and inferred RSL rate changes.

The chronology of the core for which there are PMSE estimates is derived from 3 dates (2 C14 and the inferred Hg dated horizon) spanning nearly 300 years of accumulation. This fact, coupled with the extremely low accumulation rate of saltmarshes in Greenland, makes it extremely challenging to produce 'high resolution' records. Consequently, whilst I think we can have some confidence in the century scale trends identified in this sequence, inferring decadal scale changes is much more equivocal. For example, I find it difficult to see the acceleration in the rate of RSL fall since the 1990s referred to in the text (Ln448-449). This is not evident in Fig 3C and in the absence of age control for the 20th century, it is impossible to exclude changes in accumulation rate as a cause for any reconstructed PMSE variation. Unless the authors can provide additional supporting evidence to confirm sub-century scale changes I think these would be best excluded.

We acknowledge the fact that the chronology for the core is not well constrained in the top 4 cm (since ~1850-1900 CE). We agree therefore that confidence in changes in RSL rate during this period are harder to justify. We are confident that the core provides evidence for stable and then falling RSL during the 20th Century from the diatom and LOI data but we agree that the decadal-scale increase since the 1990s is less clear. We have therefore removed reference to this acceleration in RSL fall in the text and figures. The main take home message is that the RSL data indicates a small amount of RSL fall since the end of the Little Ice Age, but the modelling suggests a much larger amount. Making the proposed change has not negated this major conclusion of the study.

I also note that in Fig 3C there is a rapid jump in the apparent rate of RSL fall for the last (topmost) sample. Does the PMSE of this sample come from the observed core top height or is it based on the diatom assemblage? The reason for asking is that if it is based on the former it is possible to introduce a spurious jump in the PMSE due to a vertical offset between the mid-point of proxy-based PMSE estimates (at least when using proxies such as foraminifera which have edge effects toward the upper limit of marine influence). When calculating rates, it may be better to be consistent and use the same (proxy-based) PMSE estimate as the rest of the core. The 'true' elevation of the marsh surface should then be found within the error bars of the PMSE estimate.

We thank Dr Edwards for raising this issue. The elevation of the uppermost sample in the core is taken from the core top height, not the predicted elevation based on the diatom assemblage. We agree that a better approach is to use the PMSE estimate based on the diatom assemblage to be consistent with the proxy-based reconstructions in the rest of the core. We have modified Figures 3-6 to this effect, and have added a sentence to the methods section to explain this approach.

**Technical Comments**

Ln 148: can bedrock 'prograde'? Alternative term may be better.

We agree this word is wrongly used here. We have replaced 'shallow prograding bedrock' with 'a shallow bedrock rise'.

Ln 363: Pinnularia intermedia in italics

We have edited this in the revised text.

Ln 430: error term for +0.2 mm/yr?

We apologise that this error term was missed from the original paper. We have added an error term of  $\pm 0.05$  mm/yr and also correction of  $\pm 0.02$ mm/yr instead of 0.08mm/yr to 1900-present day trend.

Table 1: This is confusing - what is the CE/BP? Shouldn't this be all BP (or CE / BCE)? Values in the table don't seem to match the plot in Fig 4. Perhaps it would be better to have the reported C14 age and uncertainty in yrs BP and then a separate column for the calibrated output as CE?

We agree that some of the data in Table 1 is confusing. We have modified Table 1 so that all 14C ages are reported as Modern or in years BP, and have added a separate column for uncalibrated output in CE.

Figure 3B: were diatoms analysed from the basal freshwater unit? If so (and they were absent) a note to that effect somewhere would be useful (or annotate on the figure 'barren')

The basal freshwater unit in Figure 3B was devoid of diatoms. We have added 'no diatoms present' to the figure to make this clearer.

Table S2: Depth error (6.25). Labelling errors in the lower 3 samples? 'intermedia'. Should this just be PI?

We note the typo in the depth of 6.25 cm. We also agree that where 'Intermedia' is mentioned it should be just 'PI' in this table. We have modified this in a revised version of Table S2.
* * *
**Udita Mukherjee: General Comments**

Woodroffe et al. reconstruct the relative sea level history for the last 300 years from saltmarsh sediments from the Dronning Marie Dal, near the Greenland Ice Sheet margin, and the comparison between the observed and modeled RSL during this time show a considerable mismatch. The RSL record constructed from detailed analysis of the saltmarsh sediment, which takes into account the relevant uncertainties associated with such a record, from an area that is not very well studied is an important contribution of this study. Constraining the rate and amount of RSL change as a result of the melting of GrIS is an important part of both our geological understanding as well as for implications for coastal vulnerability. This paper does a very thorough job of categorically piecing out all the variables associated with RSL change in this area, and ultimately quantifying the mismatch between the modeled and observed RSL. I recommend that this paper be published with some minor revisions.

We thank Dr Mukherjee for her positive and constructive comments on the paper, and respond to her comments below in red.

The earth model parameter space used for the GIA models and the contributions from various glaciers are standard and understandable. However, it might be interesting to add a figure that supports the statement made in line 421 "Any chosen Earth configuration within the parameter range explored does not significantly affect the predicted sea-level change...", as the solid Earth adjustment is one of the biggest contributors to the RSL balance here.

Thank you for this comment- it is correct that solid Earth adjustment is the biggest contributor. The statement is already supported by the ranges of sea level fall quoted in the subsequent lines (lines 422-423). It is difficult to see how another figure could provide more information than this. However, the text may be expanded to include the following:

"Using a fixed lithospheric thickness of 96km, the modelled total sea level fall arising from post-LIA mass loss across a suite of earth models with upper mantle viscosities ranging from  $5 \times 10^{19} - 1 \times 10^{22}$  Pa s and lower mantle viscosities in the range of  $1 \times 10^{21} - 5 \times 10^{22}$  Pa s was 0.65 to 0.86m, a difference of 0.21m which is within the uncertainty range of the RSL reconstruction (Figure 4B). The upper mantle viscosity is the largest contribution to this uncertainty accounting for both upper and lower bounds of this range. The effect of reducing the lithospheric thickness from 120km to 46km reduces the amount of modelled relative sea level fall by only a few cm."

We have added the above text into the manuscript at line 421.

I would also like to point out that most figures in this paper can be made more accessible to the reader. For example, it is quite difficult to read the names of the location in Figure 1. Figures 6a and b are difficult to read, probably arising from the use of different color scales. The authors might want to explore a different way to visualize the results of this figure, if possible.

Thank you for noticing this. We have amended Figure 1 to make the text clearer, and have edited Figures 6a/b to include a more accessible colour palette (e.g. for colourblind readers) and a consistent scale on both figures.

Fig 4c- The y-axis reads RSL (mm/yr). It should be the rate of RSL change (mm/yr).

We have edited this axis label in the revised Figure 4c.

The usage of years (BP), years (CE), and date (CE) in the paper gets confusing at times.

We have standardised the language used to CE and BCE and years CE in the text, tables and figures in the revised version of the manuscript.

Line 299-300 – The sentence "Dronning Marie Dal is proximal to..." seems to have a typo and does not make sense.

"Dronning Marie Dal is proximal to glacier sources in Iceland and Baffin Bay so should display some level of sensitivity to ice loss distribution over these glacierised areas..."

We believe that this sentence reads ok and therefore have made no change to the text here.